# Neural Model Checking

**Mirco Giacobbe**[*]
University of Birmingham, UK

**Daniel Kroening**[*]
Amazon Web Services, USA

**Abhinandan Pal**[*]
University of Birmingham, UK

**Michael Tautschnig**[*]
Amazon Web Services, USA and
Queen Mary University of London, UK

## Abstract

We introduce a machine learning approach to model checking temporal logic, with application to formal hardware verification. Model checking answers the question of whether every execution of a given system satisfies a desired temporal logic specification. Unlike testing, model checking provides formal guarantees. Its application is expected standard in silicon design and the EDA industry has invested decades into the development of performant symbolic model checking algorithms. Our new approach combines machine learning and symbolic reasoning by using neural networks as formal proof certificates for linear temporal logic. We train our neural certificates from randomly generated executions of the system and we then symbolically check their validity using satisfiability solving which, upon the affirmative answer, establishes that the system provably satisfies the specification. We leverage the expressive power of neural networks to represent proof certificates as well the fact that checking a certificate is much simpler than finding one. As a result, our machine learning procedure for model checking is entirely unsupervised, formally sound, and practically effective. We experimentally demonstrate that our method outperforms the state-of-the-art academic and commercial model checkers on a set of standard hardware designs written in SystemVerilog.

## 1 Introduction

Electronic design is complex and prone to error. Hardware bugs are permanent after production and as such can irremediably affect the correctness of software—which runs on hardware—and can compromise the safety of cyber-physical systems—which embed hardware. Correctness assurance is core to the engineering of digital circuitry, with the median FPGA and IC/ASIC projects spending respectively $40\%$ and $60\%$ of time in verification [48]. Verification approaches based on directed or constrained random testing are easy to set up but are inherently non-exhaustive [89, 91]. Testing cannot show the absence of bugs which, for systems the safety of which is critical, can have serious consequences; notably, over $40\%$ of hardware development projects must satisfy at least one functional safety standard [48]. In contrast to testing, *model checking* a design against a formal specification of correctness answers the question of whether the design satisfies the specification with mathematical certainty, for every possible execution of the system [9, 13, 35].

The EDA industry has heavily invested in software tools for symbolic model checking. Early symbolic model checking algorithms utilise fixed-point computations with binary decision diagrams (BDDs) [7], where each node specifies the Boolean assignment for a circuit's flip-flop or input bit [26, 45]. BDDs struggle to scale when applied to complex arithmetic data paths, prompting a shift towards iterative approximation of fixed points using propositional satisfiability (SAT) solving [16, 17, 33], which

---

[*]The authors are listed alphabetically.

38th Conference on Neural Information Processing Systems (NeurIPS 2024).

is now the state-of-the-art technique. Both BDD and SAT-based model checking, despite extensive research, remain computationally demanding; even small circuit modules can require days to verify or may not complete at all. Consequently, verification engineers often limit state space exploration to a bounded time horizon through bounded model checking, sacrificing global correctness over the unbounded time domain.

We present a machine learning approach to hardware model checking that leverages neural networks to represent proof certificates for the compliance of a given hardware design with a given linear temporal logic (LTL) specification [82]. Our approach avoids fixed-point algorithms entirely, capitalises on the efficient word-level reasoning of satisfiability solvers, and delivers a formal guarantee over an unbounded time horizon. Given a hardware design and an LTL specification $\Phi$, we train a word-level neural certificate for the compliance of the design with the specification from test executions, which we then check using a satisfiability solver. We leverage the observation that checking a proof certificate is much simpler than solving the model checking problem directly, and that neural networks are an effective representation of proof certificates for the correctness of systems [28, 50]. We ultimately obtain a machine learning procedure for hardware model checking that is entirely unsupervised, formally sound and, as our experiments show, very effective in practice.

Our learn-and-check procedure begins by generating a synthetic dataset through random executions of the system alongside a Büchi automaton that identifies counterexamples to $\Phi$. We then train a *neural ranking function* designed to strictly decrease whenever the automaton encounters an accepting state and remain stable on non-accepting states. After training, we formally check that the ranking function generalises to all possible executions. We frame the check as a cost-effective one-step bounded model checking problem involving the system, the automaton, and the quantised neural ranking function, which we delegate to a satisfiability solver. As the ranking function cannot decrease indefinitely, this confirms that the automaton cannot accept any system execution, effectively proving that such executions are impossible. Hence, if the solver concludes that no counterexample exists, it demonstrates that no execution satisfies $\neg\Phi$, thereby affirming that the system satisfies $\Phi$ [37, 95].

We have built a prototype that integrates PyTorch, the bounded model checker EBMC, the LTL-to-automata translator Spot, the SystemVerilog simulator Verilator, and the satisfiability solver Bitwuzla [44, 76, 80, 88]. We have assessed the effectiveness of our method across 194 standard hardware model checking problems written in SystemVerilog and compared our results with the state-of-the-art academic hardware model checkers ABC and nuXmv [24, 27], and two commercial counterparts. For any given time budget of less than 5 hours, our method completes on average $60\,\%$ more tasks than ABC, $34\,\%$ more tasks than nuXmv, and $11\,\%$ more tasks than the leading commercial model checker. Our method is faster than the academic tools on $67\,\%$ of the tasks, 10X faster on $34\,\%$, and 100X faster on $4\,\%$; when considering the leading commercial tool, our method is faster on $75\,\%$, 10X faster on $29\,\%$, and 100X faster on $2\,\%$ of them. Overall, with a straightforward implementation, our method outperforms mature academic and commercial model checkers.

Our contribution is threefold. We present for the first time a hardware model checking approach based on neural certificates. We extend neural ranking functions, previously introduced for the termination analysis of software, to LTL model checking and the verification of reactive systems. We have built a prototype and experimentally demonstrated that our approach compares favourably with the leading academic and commercial hardware model checkers. Our technology delivers formal guarantees of correctness and positively contributes to the safety assurance of systems.

## 2   Automata-theoretic Linear Temporal Logic Model Checking

An LTL model checking problem consists of a model $\mathcal{M}$ that describes a system design and an LTL formula $\Phi$ that describes the desired temporal behaviour of the system [52, 82]. The problem is to decide whether all traces of $\mathcal{M}$ satisfy $\Phi$.

Our formal model $\mathcal{M}$ of a hardware design consists of a finite set of bit-vector-typed variables $X_{\mathcal{M}}$ with fixed bit-width and domain of assignments $S$, partitioned into input variables $\mathrm{inp}\,X_{\mathcal{M}} \subseteq X_{\mathcal{M}}$ and state-holding register variables $\mathrm{reg}\,X_{\mathcal{M}} \subseteq X_{\mathcal{M}}$; we interpret primed variables $X'_{\mathcal{M}}$ as the value of $X_{\mathcal{M}}$ after one clock cycle. Then, a sequential update relation $\mathrm{Update}_{\mathcal{M}}$ relates $X_{\mathcal{M}}$ and $\mathrm{reg}\,X'_{\mathcal{M}}$ and computes the next-state valuation of the registers from the current-state valuation of all variables; we interpret $\mathrm{Update}_{\mathcal{M}}$ as a first-order logic formula encoding this relation. A state $s \in S$ is a valuation for the variables $X_{\mathcal{M}}$. We denote as $\mathrm{reg}\,s, \mathrm{inp}\,s, \ldots$ the restriction of $s$ to the respective

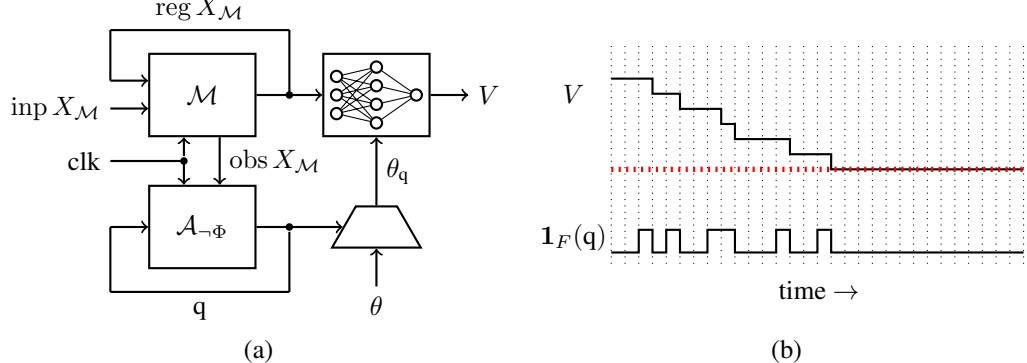

Figure 1: Automata-theoretic neural model checking via fair termination

class of variables. For two states $s$ and $s'$, the state $s'$ is a successor of $s$, which we write as $s \rightarrow_{\mathcal{M}} s'$, if $\mathrm{Update}_{\mathcal{M}}(s, \mathrm{reg}\, s')$ evaluates to true. We call $\rightarrow_{\mathcal{M}}$ the transition relation of $\mathcal{M}$ and say that an infinite sequence of states $\bar{s}_0, \bar{s}_1, \bar{s}_2, \ldots$ is an execution of $\mathcal{M}$ if $\bar{s}_i \rightarrow_{\mathcal{M}} \bar{s}_{i+1}$ for all $i \geq 0$; we say that an execution is initialised in $s_0 \in S$ when $\bar{s}_0 = s_0$.

We specify the intended behaviour of systems in LTL, which is the foundation of SystemVerilog Assertions. LTL extends propositional logic with temporal modalities X, G, F, and U. The modality X $\Phi_1$ indicates that $\Phi_1$ holds immediately after one step in the future, G $\Phi_1$ indicates that $\Phi_1$ holds at all times in the future, F $\Phi_1$ indicates that $\Phi_1$ holds at some time in the future, and $\Phi_1$ U $\Phi_2$ indicates that $\Phi_1$ holds at all times until $\Phi_2$ holds at some time in the future. We refer the reader to the literature for the formal syntax and semantics of LTL [82]. The atomic propositions of the LTL formulae we consider are Boolean variables of $\mathcal{M}$, which we call the observables $\mathrm{obs}\, X_{\mathcal{M}} \subseteq X_{\mathcal{M}}$ of $\mathcal{M}$. We note that any first-order predicate over $X_{\mathcal{M}}$ can be bound to a Boolean observable using combinational logic (cf. Figure 4, where observable `ful` corresponds to predicate `cnt == 7`).

We call a trace of $\mathcal{M}$ a sequence $\mathrm{obs}\, \bar{s}_0, \mathrm{obs}\, \bar{s}_1, \mathrm{obs}\, \bar{s}_2, \ldots$ where $\bar{s}_0, \bar{s}_1, \bar{s}_2, \ldots$ is an execution of $\mathcal{M}$. We define the language $L_{\mathcal{M}}$ of $\mathcal{M}$ as the maximal set of traces of $\mathcal{M}$. Every LTL formula $\Phi$ is interpreted over traces and as such defines the language $L_{\Phi}$ of traces that satisfy $\Phi$. The model checking problem corresponds to deciding the language inclusion question $L_{\mathcal{M}} \subseteq L_{\Phi}$.

As is standard in automata-theoretic model checking, we rely on the result that every LTL formula admits a non-deterministic Büchi automaton that recognises the same language [95, 96]. A non-deterministic Büchi automaton $\mathcal{A}$ consists of a finite set of states $Q$, an initial start state $q_0 \in Q$, an input domain $\Sigma$ (also called alphabet), a transition relation $\delta \subseteq Q \times \Sigma \times Q$, and a set of fair states $F \subseteq Q$. One can interpret an automaton $\mathcal{A}$ as a hardware design with one register variable $\mathrm{reg}\, X_{\mathcal{A}} = \{q\}$ having domain $Q$, input and observable variables $\mathrm{inp}\, X_{\mathcal{A}} = \mathrm{obs}\, X_{\mathcal{A}}$ having domain $\Sigma$, and sequential update relation $\mathrm{Update}_{\mathcal{A}}(\sigma, q, q') \equiv (q, \sigma, q') \in \delta$ governing the automaton state transitions. We say that an execution of $\mathcal{A}$ is *fair* (also said to be an accepting execution) if it visits fair states infinitely often. We define the fair language $L^{\mathrm{f}}_{\mathcal{A}}$ of $\mathcal{A}$ as the maximal set of traces corresponding to fair executions initialised in $q_0$. Given any LTL formula $\Phi$, there are translation algorithms and tools to construct non-deterministic Büchi automata $\mathcal{A}_{\Phi}$ such that $L^{\mathrm{f}}_{\mathcal{A}_{\Phi}} = L_{\Phi}$ [44, 58].

The standard approach to answer the language inclusion question $L_{\mathcal{M}} \subseteq L_{\Phi}$ is to answer the dual language emptiness question $L_{\mathcal{M}} \cap L_{\neg\Phi} = \emptyset$ [13, 35]. For this purpose, we first construct a non-deterministic Büchi automaton $\mathcal{A}_{\neg\Phi}$ for the complement specification $\neg\Phi$ where $\mathrm{inp}\, X_{\mathcal{A}_{\neg\Phi}} = \mathrm{obs}\, X_{\mathcal{M}}$, then we reason over the synchronous composition (over a shared clock) of $\mathcal{M}$ and $\mathcal{A}_{\neg\Phi}$ as illustrated in Figure 1a. We direct the reader to the relevant literature for general definitions of system composition [10]. In this context, the synchronous composition results in the system $\mathcal{M} \parallel \mathcal{A}_{\neg\Phi}$ with input variables $\mathrm{inp}\, X_{\mathcal{M} \parallel \mathcal{A}_{\neg\Phi}} = \mathrm{inp}\, X_{\mathcal{M}}$, register variables $\mathrm{reg}\, X_{\mathcal{M} \parallel \mathcal{A}_{\neg\Phi}} = \mathrm{reg}\, X_{\mathcal{M}} \cup \{q\}$, observable variables $\mathrm{obs}\, X_{\mathcal{M} \parallel \mathcal{A}_{\neg\Phi}} = \mathrm{obs}\, X_{\mathcal{M}}$, and sequential update relation $\mathrm{Update}_{\mathcal{M} \parallel \mathcal{A}_{\neg\Phi}}(s, q, r', q') = \mathrm{Update}_{\mathcal{M}}(s, r') \wedge \mathrm{Update}_{\mathcal{A}_{\neg\Phi}}(\mathrm{obs}\, s, q, q')$. We extend the fair states of $\mathcal{A}_{\neg\Phi}$ to $\mathcal{M} \parallel \mathcal{A}_{\neg\Phi}$, i.e., we define them as $\{(s, q) \mid s \in S, q \in F\}$, and as a result we have that $L^{\mathrm{f}}_{\mathcal{M} \parallel \mathcal{A}_{\neg\Phi}} = L_{\mathcal{M}} \cap L^{\mathrm{f}}_{\mathcal{A}_{\neg\Phi}} = L_{\mathcal{M}} \cap L_{\neg\Phi}$. This reduces our language emptiness question to the equivalent *fair emptiness* problem $L^{\mathrm{f}}_{\mathcal{M} \parallel \mathcal{A}_{\neg\Phi}} = \emptyset$.

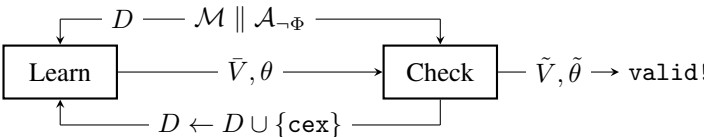

Figure 2: Learn-and-check workflow for *provably sound* neural ranking function learning

The fair emptiness problem amounts to showing that all executions of $\mathcal{M} \parallel \mathcal{A}_{\neg\Phi}$ are unfair, and we do so by presenting a ranking function that witnesses fair termination [51, 67]. A ranking function for fair termination is a map $V \colon \operatorname{reg} S \times Q \to R$ where $(R, \prec)$ defines a well-founded relation and, for all system and automaton states $\boldsymbol{s}, \boldsymbol{s}' \in S, \boldsymbol{q}, \boldsymbol{q}' \in Q$, the following two conditions hold true:

$$(\boldsymbol{s}, \boldsymbol{q}) \to_{\mathcal{M}\|\mathcal{A}_{\neg\Phi}} (\boldsymbol{s}', \boldsymbol{q}') \implies V(\operatorname{reg} \boldsymbol{s}, \boldsymbol{q}) \succeq V(\operatorname{reg} \boldsymbol{s}', \boldsymbol{q}') \tag{1}$$

$$(\boldsymbol{s}, \boldsymbol{q}) \to_{\mathcal{M}\|\mathcal{A}_{\neg\Phi}} (\boldsymbol{s}', \boldsymbol{q}') \wedge \boldsymbol{q} \in F \implies V(\operatorname{reg} \boldsymbol{s}, \boldsymbol{q}) \succ V(\operatorname{reg} \boldsymbol{s}', \boldsymbol{q}') \tag{2}$$

A ranking function $V$ strictly decreases every time a transition from a fair state is taken, and never increases in any other case. Since every strictly decreasing sequence must be bounded from below (well-foundedness), every fair state can be visited at most finitely many times; the intuition is presented in Figure 1b, where $\mathbf{1}_F(q)$ denotes the indicator function of $F$, returning 1 if $q \in F$ and 0 otherwise. The existence of a valid ranking function represented in some form establishes that every execution of $\mathcal{M} \parallel \mathcal{A}_{\neg\Phi}$ is necessarily unfair [95]. In this work, we represent ranking functions as neural networks, the parameters of which we train from generated sample executions.

## 3 Neural Ranking Functions for Fair Termination

We approach the problem of computing a ranking function for fair termination by training a neural network $\bar{V} \colon \mathbb{R}^n \times \Theta \to \mathbb{R}$, with $n$ input neurons where $n = |\operatorname{reg} X_{\mathcal{M}}|$ is the number of register variables of the system, one output neuron, and with a space of learnable parameters $\Theta$ for its weights and biases. We associate a distinct trainable parameter $\theta_q \in \Theta$ to each state $q \in Q$ of the Büchi automaton. We train these parameters on sampled executions of $\mathcal{M} \parallel \mathcal{A}_{\neg\Phi}$ to ultimately represent a ranking function as a neural network $V(r, q) \equiv \bar{V}(r; \theta_q)$, which we call a neural ranking function. This scheme is illustrated in Figure 1, where we denote the set of all parameters by the unindexed $\theta$.

We define our training objective as fulfilling conditions (1) and (2) on our synthetic dataset of sampled executions which, by analogy with reinforcement learning, can be viewed as a special case of episodes [53, 55]. Subsequently, we verify the conditions symbolically over the full state space $S \times Q$ using satisfiability solving modulo theories (SMT) [14, 60], to confirm the validity of our neural ranking function or obtain a counterexample for re-training. Overall, our approach combines learning and SMT-based checking for both efficacy and formal soundness, as illustrated in Figure 2.

For a system $\mathcal{M}$ and a specification $\Phi$, we train the parameters $\theta$ of a neural network $\bar{V}$ from a sample dataset $D \subset \operatorname{reg} S \times Q \times \operatorname{reg} S \times Q$ of subsequent transition pairs, which we construct from random executions of the synchronous composition $\mathcal{M} \parallel \mathcal{A}_{\neg\Phi}$. Each execution $(\bar{\boldsymbol{s}}_0, \bar{\boldsymbol{q}}_0), (\bar{\boldsymbol{s}}_1, \bar{\boldsymbol{q}}_1), \ldots, (\bar{\boldsymbol{s}}_k, \bar{\boldsymbol{q}}_k)$ initiates from a random system and automaton state pair and is then simulated over a finite number of steps; the inputs to $\mathcal{M}$ and the non-deterministic choices in $\mathcal{A}_{\neg\Phi}$ are resolved randomly. Our dataset $D$ is constructed as the set of all quadruples $(\operatorname{reg} \bar{\boldsymbol{s}}_i, \bar{\boldsymbol{q}}_i, \operatorname{reg} \bar{\boldsymbol{s}}_{i+1}, \bar{\boldsymbol{q}}_{i+1})$ for $i = 0, \ldots, k-1$ from all sampled executions, capturing consecutive state pairs along each execution; notably, the order in which quadruples are stored in $D$ is immaterial for our purpose, as our method reasons and trains locally on each transition pair regardless of their order of appearance along any execution.

We train the parameters of our neural network $\bar{V}$ to satisfy the ranking function conditions (1) and (2) over $D$. For each quadruple $(\boldsymbol{r}, \boldsymbol{q}, \boldsymbol{r}', \boldsymbol{q}') \in D$, this amounts to minimising the following loss function:

$$\mathcal{L}_{\text{Rank}}(\boldsymbol{r}, \boldsymbol{q}, \boldsymbol{r}', \boldsymbol{q}'; \theta) = \operatorname{ReLU}(\bar{V}(\boldsymbol{r}'; \theta_{\boldsymbol{q}'}) - \bar{V}(\boldsymbol{r}; \theta_{\boldsymbol{q}}) + \epsilon \cdot \mathbf{1}_F(\boldsymbol{q})). \tag{3}$$

where $\epsilon > 0$ is a hyper-parameter that denotes the margin for the decrease condition. When $\mathcal{L}_{\text{Rank}}$ takes its minimum value—which is zero—then the following two cases are satisfied: if $\boldsymbol{q} \notin F$, then $\bar{V}$ does not increase along the given transition, i.e., $\bar{V}(\boldsymbol{r}; \theta_{\boldsymbol{q}}) \geq \bar{V}(\boldsymbol{r}'; \theta_{\boldsymbol{q}'})$, which corresponds to satisfy condition (1); if otherwise $\boldsymbol{q} \in F$, then $\bar{V}$ decreases by at least the margin $\epsilon > 0$ along the given transition, i.e., $\bar{V}(\boldsymbol{r}; \theta_{\boldsymbol{q}}) \geq \bar{V}(\boldsymbol{r}'; \theta_{\boldsymbol{q}'}) + \epsilon$, which corresponds to satisfy condition (2).

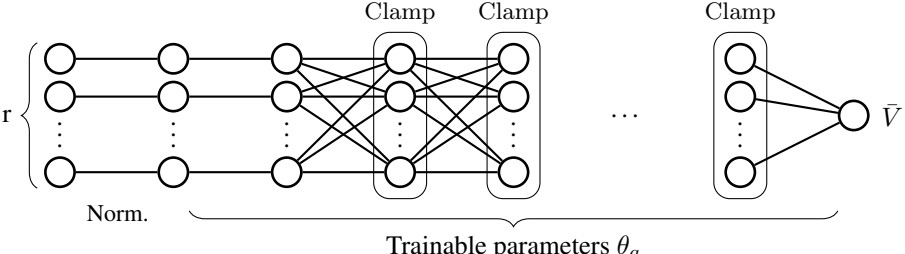

Figure 3: Neural ranking function architecture

Overall, our learning phase ensures that the total loss function $\mathcal{L}(D; \theta)$ below takes value zero:

$$\mathcal{L}(D; \theta) = \mathbb{E}_{(\boldsymbol{r}, \boldsymbol{q}, \boldsymbol{r}', \boldsymbol{q}') \in D}[\mathcal{L}_{\text{Rank}}(\boldsymbol{r}, \boldsymbol{q}, \boldsymbol{r}', \boldsymbol{q}'; \theta)] \tag{4}$$

Unlike many other machine learning applications, for our purpose it is essential to attain the global minimum; if this fails, there are counterexamples to $\bar{V}$ being a ranking function in the dataset $D$ itself. To facilitate the optimisation process, we train the parameters associated to each automaton state independently, one after the other, as opposed to training all parameters at once. Iteratively, we select one automaton state $q \in Q$ and optimise only $\theta_q \in \Theta$ for a number of steps, while keeping all other parameters $\theta_{q'} \in \Theta$ fixed to their current value, for all $q' \neq q$. We repeat the process over each automaton state, possibly iterating over the entire set of automaton states $Q$ multiple times, until the total loss $\mathcal{L}(D; \theta)$ takes value zero.

Our neural network $\bar{V}$ follows a feed-forward architecture as depicted in Figure 3: for a given automaton state $q \in Q$ and associated parameter $\theta_q$, it takes an $n$-dimensional input $r \in \mathbb{R}^n$ where each input neuron corresponds to the value of a register variable in $\operatorname{reg} X_{\mathcal{M}}$, and produces one output for the corresponding ranking value $\bar{V}(r; \theta_q)$. Our architecture consists of a normalisation layer, followed by an element-wise multiplication layer, in turn followed by a multi-layer perceptron with clamped ReLU activation functions. The first layer applies a scaling factor to each input neuron independently to ensure consistent value ranges across inputs, implemented via element-wise multiplication with a constant vector of scaling coefficients derived from the dataset $D$ before training; this integrates data normalisation into the network, enables $\bar{V}$ to use raw data from $\mathcal{M}$ and simplifies the symbolic encoding of the normalisation operation during the verification phase. The second layer applies a trainable scaling factor to each individual neuron and is implemented via element-wise multiplication with a $n$-dimensional vector with trainable coefficients. Finally, this is followed by a fully connected multi-layer perceptron with trainable weights and biases, with the activation function defined as the element-wise application of $\operatorname{Clamp}(x; u) = \max(0, \min(x, u))$; the upper bound $u$ and the depth and width of the hidden layers of the multi-layer perceptron component are hyper-parameters chosen to optimise training and verification performance.

Attaining zero total loss $\mathcal{L}(D; \theta)$ guarantees that our neural ranking function candidate $\bar{V}$ satisfies the ranking criteria for fair termination over the dataset $D$ but not necessarily over the entire transition relation $\to_{\mathcal{M} \| \mathcal{A}_{\neg \Phi}}$, as required to fulfil conditions (1) and (2) and consequently to answer our model checking question (cf. Section 2). To formally check whether the ranking criteria are satisfied over the entire transition relation, we couple our learning procedure with a sound decision procedure that verifies their validity, as illustrated in Figure 2.

We check the validity of our candidate ranking neural network using satisfiability modulo the theory of bit-vectors. While the sequential update relation $\operatorname{Update}_{\mathcal{M} \| \mathcal{A}_{\neg \Phi}}$ is natively expressed over the theory of bit-vectors, the formal semantics of the neural network $\bar{V}$ is defined on the reals. Hence, encoding $\bar{V}$ and $\operatorname{Update}_{\mathcal{M} \| \mathcal{A}_{\neg \Phi}}$ within the same query would result in a combination of real and bit-vector theories, which is supported in modern SMT solvers but often leads to sub-optimal performance [60]. Therefore, to leverage the efficacy of specialised solvers for the theory of bit-vectors [80], we quantise our neural network using a standard approach for this purpose [57]; this converts all arithmetic operations within the neural networks into fixed-point arithmetic, which are implemented using integer arithmetic only. We quantise our parameters to their respective integer representation $\tilde{\theta} \approx 2^f \cdot \theta$, where $f$ is a hyper-parameter for the number of fractional digits in fixed-point representation, and we replace linear layers and activation functions by their quantised counterpart; readers may consult the relevant literature for more detailed information on neural

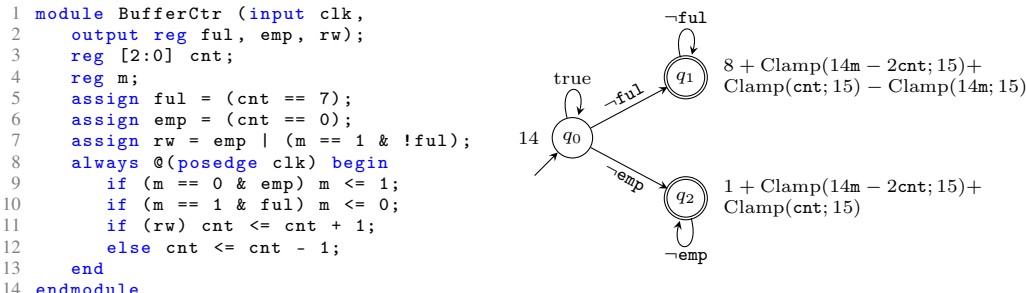

```
 1  module BufferCtr (input clk,
 2      output reg ful, emp, rw);
 3      reg [2:0] cnt;
 4      reg m;
 5      assign ful = (cnt == 7);
 6      assign emp = (cnt == 0);
 7      assign rw = emp | (m == 1 & !ful);
 8      always @(posedge clk) begin
 9          if (m == 0 & emp) m <= 1;
10          if (m == 1 & ful) m <= 0;
11          if (rw) cnt <= cnt + 1;
12          else cnt <= cnt - 1;
13      end
14  endmodule
```

Figure 4: Illustrative hardware design, Büchi automaton, and respective ranking function

network quantisation [49, 57]. This results in a quantised neural network $\tilde{V} \colon \mathbb{Z}^n \times \tilde{\Theta} \to \mathbb{Z}$ that approximates our trained network $\tilde{V} \approx 2^f \cdot \bar{V}$, where $\tilde{\Theta}$ denotes the space of integer parameters. fractional digits introduced by the linear layers [49, 57]. We consider the quantised network $\tilde{V}$ as our candidate proof certificate for fair termination.

We reduce the validity query—whether our quantised neural network $\tilde{V}$ satisfies the ranking criteria for fair termination (1) and (2) over the entire transition relation of $\mathcal{M} \parallel \mathcal{A}_{\neg\Phi}$—to the dual satisfiability query for the existence of a counterexample to the criteria. Specifically, we delegate to an off-the-shelf SMT solver the task of computing a satisfying assignment $s \in S, r' \in \mathrm{reg}\, S$ for which the following quantifier-free first-order logic formula is satisfied:

$$\bigvee_{q,q' \in Q} \mathrm{Update}_{\mathcal{M}\parallel\mathcal{A}_{\neg\Phi}}(s, q, r', q') \wedge \tilde{V}(\mathrm{reg}\, s; \tilde{\boldsymbol{\theta}}_q) - \mathbf{1}_F(q) < \tilde{V}(r'; \tilde{\boldsymbol{\theta}}_{q'}) \tag{5}$$

where $\tilde{\boldsymbol{\theta}}$ is the (constant) parameter resulting from training and quantisation. We encode the quantised neural network $\tilde{V}$ using a standard translation into first-order logic over the theory of bit-vectors [49], supplementing it with specialised rewriting rules to enhance the solver's performance, as detailed in Appendix A. We additionally note that $\tilde{V}$ is guaranteed to be bounded from below as $S$ is finite, albeit potentially very large, i.e., exponential in the combined bit-width of $X_{\mathcal{M}}$.

If the solver finds a satisfying assignment, then the assignment represents a transition of $\mathcal{M}$ that refutes the validity of $\tilde{V}$; in this case, we extend it to a respective transition in $\mathcal{M} \parallel \mathcal{A}_{\neg\Phi}$, we add it to our dataset $D$ and repeat training and verification in a loop. Conversely, if the solver determines that formula (5) is unsatisfiable, then our procedure concludes that $\tilde{V}$ is formally a valid neural ranking function and, consequently, system $\mathcal{M}$ satisfies specification $\Phi$.

We note that LTL model checking of hardware designs is decidable and PSPACE-complete [9, 13, 35]. While it is theoretically possible for our approach to achieve completeness when a ranking function exists by enumerating all transitions and employing a sufficiently large neural network as a lookup table over the entire state space, this is impractical for all but toy cases. In this work, we employ tiny neural networks and incomplete but practically effective gradient descent algorithms to train neural ranking functions. We experimentally demonstrate on a large set of formal hardware verification benchmarks that this solution is very effective in practice.

## 4   Illustrative Example

Modern hardware designs frequently incorporate word-level arithmetic operations, the simplest of which being counter increments/decrements, which are a staple in hardware engineering [71, 98]. One such example is illustrated as part of the SystemVerilog module in Figure 4. This represents a simplified buffer controller that counts the number of packets stored in the buffer and indicates when the buffer is full or empty with the `ful` and `emp` signals, respectively. This specific controller internally coordinates read-and-write operations through the `rw` signal: iteratively, the system signals `rw = 1` until the buffer is full and then `rw = 0` until the buffer is empty.

The design satisfies the property that both our observables `ful` and `emp` are true infinitely often, captured by the LTL formula $\Phi = \mathsf{GF}\, \mathtt{ful} \wedge \mathsf{GF}\, \mathtt{emp}$. Dually, this specification says that the system

does not eventually go into a state from where ¬`ful` holds indefinitely nor ¬`emp` holds indefinitely, that is, ¬Φ = FG ¬`ful` ∨ FG ¬`emp`. Equivalently, this amounts to proving that no system trace is in the fair language of the automaton $\mathcal{A}_{\neg\Phi}$ given in Figure 4.

A neural ranking function $\bar{V}$ for the fair termination of this system and automaton has 5 input neurons for the register variables `cnt`, `m`, `ful`, `emp`, and `rw`, and one hidden layer with three neurons in the multi-layer perceptron component. As illustrated in Figure 4, each automaton state is associated with a ranking function defined in terms of this architecture and their respective parameters. The sequence below gives an execution of model states alongside the respective ranking function values:

| | emp | | | | | | | ful | | | | | | | emp | | | | | | | ful |
|---|---|---|---|---|---|---|---|---|---|---|---|---|---|---|---|---|---|---|---|---|---|---|
| `cnt` | 0 | 1 | 2 | 3 | 4 | 5 | 6 | 7 | 6 | 5 | 4 | 3 | 2 | 1 | 0 | 1 | 2 | 3 | 4 | 5 | 6 | 7 |
| `m` | 0 | 1 | 1 | 1 | 1 | 1 | 1 | 1 | 0 | 0 | 0 | 0 | 0 | 0 | 0 | 1 | 1 | 1 | 1 | 1 | 1 | 1 |
| `rw` | 1 | 1 | 1 | 1 | 1 | 1 | 1 | 0 | 0 | 0 | 0 | 0 | 0 | 0 | 1 | 1 | 1 | 1 | 1 | 1 | 1 | 0 |
| $\bar{V}(\cdot;\theta_{q_0})$ | 14 | 14 | 14 | 14 | 14 | 14 | 14 | 14 | 14 | 14 | 14 | 14 | 14 | 14 | 14 | 14 | 14 | 14 | 14 | 14 | 14 | 14 |
| $\bar{V}(\cdot;\theta_{q_1})$ | 8 | 7 | 6 | 5 | 4 | 3 | 2 | 1 | 14 | 13 | 12 | 11 | 10 | 9 | 8 | 7 | 6 | 5 | 4 | 3 | 2 | 1 |
| $\bar{V}(\cdot;\theta_{q_2})$ | 1 | 14 | 13 | 12 | 11 | 10 | 9 | 8 | 7 | 6 | 5 | 4 | 3 | 2 | 1 | 14 | 13 | 12 | 11 | 10 | 9 | 8 |

One can observe that all transitions throughout this execution satisfy conditions (1) and (2). This assessment is based on the (not explicitly presented) synchronous composition with the automaton. First, we note that every transition originating from $q_0$ has a non-increasing ranking value, as $\bar{V}(\cdot;\theta_{q_0}) = 14$ is an upper bound to all other values. Furthermore, every transition leaving $q_1$—that is, every transition whose source state satisfies ¬`ful`—exhibits a strictly decreasing value $\bar{V}(\cdot;\theta_{q_1})$. Similarly, the same observation applies to $q_2$ and the condition ¬`emp`. We note that the transitions that exhibit increasing values from 1 to 14 in this execution are impossible over the synchronous composition; this is because they are originating from states that satisfy both `ful` and $q_1$, and similarly states that satisfy both `emp` and $q_2$, and which do not have corresponding transitions in the automaton.

This neural ranking function admits no increasing transition originating from $q_0$ and no non-decreasing transitions originating from $q_1$ or $q_2$ on the synchronous composition of the system and the automaton. Therefore, it is a valid proof certificate for every system trace to satisfy specification Φ.

## 5 Experimental Evaluation

We examine 194 verification tasks derived from ten parameterised hardware designs, detailed in Appendix B. By adjusting parameter values, we create tasks of varying complexity, resulting in different logic gate counts and state space sizes, thus offering a broad spectrum of verification complexity for tool comparison. The parameter ranges for each design are given as "all tasks" in Figure 5. These tasks serve as benchmarks to evaluate the scalability of our method relative to conventional model checking.

**Implementation**   We have developed a prototype tool for neural model checking[2], utilising Spot 2.11.6 [44] to generate the automaton $\mathcal{A}_{\neg\Phi}$ from an LTL specification Φ. As depicted in Figure 1, the circuit model $\mathcal{M}$ and the automaton $\mathcal{A}_{\neg\Phi}$ synchronise over a shared clock to form a product machine. Using Verilator version 5.022 [88], we generate a dataset $D$ from finite trajectories of this machine. This dataset trains a neural network using PyTorch 2.2.2, as outlined in Section 3. To ensure formal guarantees, the network is quantised and subsequently translated to SMT, following the process outlined in Appendix A. The SystemVerilog model is converted to SMT using EBMC 5.2 [76]. We check the satisfiability problem using the Bitwuzla 0.6.0 SMT solver [80].

**State of the Art**   We benchmarked our neural model checking approach against two leading model checkers, nuXmv [27] and ABC [24, 25]. ABC and nuXmv were the top performers in the liveness category of the hardware model checking competition (HWMCC) [15, 19]. Our comparison employed the latest versions: nuXmv 2.0.0 and ABC's Super Prove tool suite [25], which were also used in the most recent HWMCC'20 [15]. We further consider two widely used industrial formal verification tools for SystemVerilog, anonymised as industry tool X and industry tool Y. Tool Y fails to complete any of the 194 tasks and is therefore not referenced further in this section.

---

[2]`https://github.com/aiverification/neuralmc`

Table 1: Number of verification task completed by academic and industrial tool, per design

|  | LS | LCD | Tmcp | i2cS | 7-Seg | PWM | VGA | UARTt | Delay | Gray | Total |
|---|---|---|---|---|---|---|---|---|---|---|---|
| Tasks | 16 | 14 | 17 | 20 | 30 | 12 | 10 | 10 | 32 | 33 | 194 |
| ABC | 2 | 3 | 7 | 3 | 8 | 2 | 3 | **10** | 6 | 13 | 57 |
| nuXmv | 8 | 9 | 12 | 10 | 10 | 7 | 3 | **10** | 24 | 24 | 117 |
| our | 15 | **14** | **17** | **18** | **30** | 11 | 0 | **10** | **32** | **33** | 180 |
| Ind. X | **16** | **14** | **17** | **18** | 18 | **12** | **10** | **10** | 19 | 22 | 156 |
| Ind. Y | 0 | 0 | 0 | 0 | 0 | 0 | 0 | 0 | 0 | 0 | 0 |

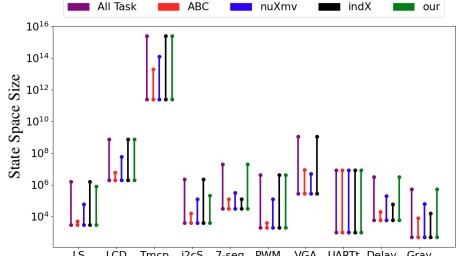 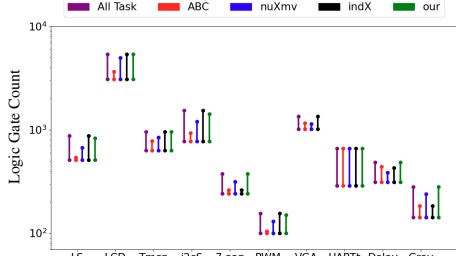

Figure 5: Solved tasks in terms of state space size and logic gate count (log scale)

**Experimental Setup**  Evaluations were conducted on an Intel Xeon 2.5 GHz processor with eight threads and 32 GB of RAM running Ubuntu 20.04. Bitwuzla and nuXmv utilise one core each, ABC used three cores, and PyTorch leveraged all available cores. Each tool was allotted a maximum of five hours for each verification task, as detailed in Appendix C.

**Hyper-parameters**  We instantiate the architecture described in Section 3 and illustrated in Figure 3, employing two hidden layers containing 8 and 5 neurons. The normalisation layer scales the input values to the range [0, 100]. We train with the AdamW optimiser [70], typically setting the learning rate to 0.1 or selecting from 0.2, 0.05, 0.03, 0.01 if adjusted, with a fixed weight decay of 0.01, demonstrating minimal hyperparameter tuning for training.

**Dataset Generation**  In hardware design, engineers utilise test benches to verify safety properties through directed testing or Constraint Random Verification (CRV), aiming for high coverage and capturing edge cases [48, 89]. We apply CRV to the SystemVerilog file, generating random trajectories. As outlined in Section 3, we start these trajectories by selecting the internal states of model $\mathcal{M}$ (e.g., `module BufferCtr` and automaton $A_{\neg\Phi}$; in Figure 4) using a uniform distribution. At each step, we assign random inputs to model $\mathcal{M}$ and handle the non-determinism in automaton $A_{\neg\Phi}$ by making choices from uniform or skewed distributions. We skew the distribution when a particular event is too predominant or too rare. In our experiments, such skewing is rare and limited to the reset and enable signals in $\mathcal{M}$, as well as the non-determinism in the automaton $A_{\neg\Phi}$.

**Solved Tasks**  Table 1 presents the number of completed tasks for each tool across the ten hardware designs, while Figure 5 shows the range of state-space sizes and logic gate counts each tool successfully handled. Overall, our tool performs favourably in comparison to others, with the notable exception of the VGA design, where training a ranking function failed due to local minima, preventing convergence to zero loss—a known limitation of gradient descent-based methods.

**Aggregate Runtime Comparison**  Figure 6a displays a cactus plot with a 5 h limit, we consider our configuration with 8 and 5 hidden neurons as detailed in the section, along with the aggregate of the best time on individual tasks obtained from our ablation study, as detailed in Appendix D. While the default architecture performs the best across all tasks, on some tasks a smaller network is sufficient, and leads to lower verification time. At the same time, larger networks often succeed on tasks that otherwise fail, making the "our best" line strictly better than "our (5, 8)". This shows that improvement can be obtained by tuning the width of the hidden layers; note that this analysis considers three additional configurations (i.e, (3, 2), (5, 3), (15, 8)) that adhere to the architecture introduced in Section 3. For the rest of our experiments, we continue using the default architecture.

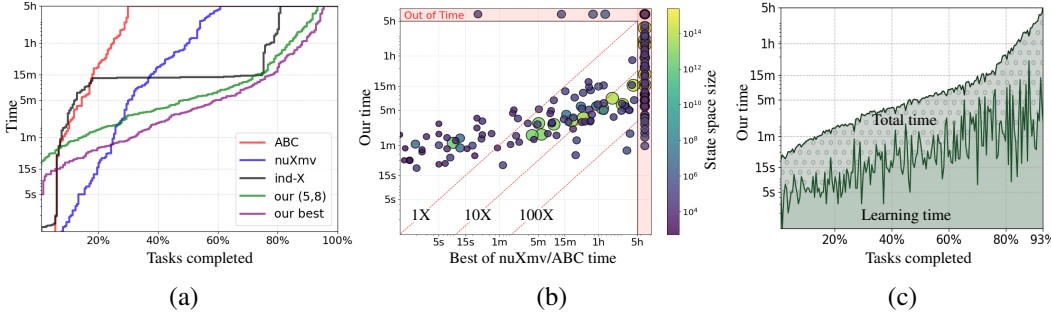

<p align="center">(a)             (b)             (c)</p>

Figure 6: Runtime comparison with the state of the art (all times are in log scale)

The plot further shows that our tool completes 93 % of tasks, outperforming ABC, nuXmv, and industry tool X, which completes 29 %, 60 %, and 80 %, respectively. At any point in the time axis, we compute the difference between the percentage of tasks completed by our tool with each of the others in the figure. Then, taking the average of these differences across the time axis, showing that our method is successful in 60 % more tasks than ABC, 34 % more than nuXmv, and 11 % more than the leading commercial model checker at any given time. Furthermore, the number of tasks completed by nuXmv in 5 h are finished by our tool in less than 8 min, and those completed by ABC in 5 h take just under 3 min with our method.

**Individual Runtime Comparison**  Figure 6b presents a scatter plot where each point represents a verification task, with size and brightness indicating the state-space size. Points are plotted horizontally by the lesser of time taken by nuXmv or ABC and vertically by our method's time. The plot reveals that academic tools time out on 39 % of tasks, while our method times out on 7 %. Moreover, we are faster than the academic tools on 67 % of tasks, 10 times faster on 34 %, and 100 times faster on 4 %. These results demonstrate that we generally outperform the state of the art on this benchmark set (see Appendix 3 for individual runtimes). However, we perform relatively worse on the `UARTt` design. This design involves an $N$-bit register for data storage and a counter for transmitted bits, enabling sequential outputs. Since there is no word-level arithmetic over the $N$-bit register, increasing its size minimally affects the complexity of symbolic model checking. Consequently, ABC, nuXmv, and industry tool X complete all `UARTt` tasks in under a second, while our tool takes a few minutes due to overhead from the sampling, learning, and SMT-check steps, making us slower on trivial model-checking problems.

**Learning vs. Checking Time**  Figure 6c illustrates the time split between learning the neural network—which involves dataset generation and training—and verifying it as a valid ranking function. The lower line indicates learning time; the upper line represents total time, with the gap showing the time spent on SMT checking. Extensive sampling across a broad range of trajectories covering most edge cases led our method to learn the network directly without needing retraining due to counterexamples in the SMT-check phase, except in four tasks. The plot shows that 93 % of tasks were trained successfully, generally within five minutes, and remarkably, the 70 % were completed in under a minute. For tasks that did not train to zero loss, the 5 h time limit was not fully utilised; the loss function stabilised at local minima in just a few minutes. Moreover, training was faster than verification on 97 % tasks—10 times faster on 46 % and 100 times faster on 6 %.

**Limitations**  The primary limitation of our approach arises from the extended SMT-check times and the risk of getting trapped in local minima. Despite these challenges, our method consistently outperforms traditional symbolic model checkers while relying on off-the-shelf SMT solvers and machine learning optimisers. Additionally, our neural architecture requires numerical inputs at the word level, which limits its application to bit-level netlists. This limitation is not high-impact, as modern formal verification tools predominantly utilise Verilog RTL rather than netlist representations.

**Threats to Validity**  The experimental results may not generalise to other workloads. As any work that relies on benchmarks, our benchmarks may not be representative for other workloads. We mitigate this threat by selecting extremely common hardware design patterns from the standard literature. We remark that our data sets we use to train the neural nets do not suffer from the common threat of training data bias, and the common out-of-distribution problem: we train our neural net from scratch for each benchmark using randomly generated trajectories, and do not use any pretraining.

# 6 Related Work

Formal verification, temporal logic and model checking have been developed for more than fifty years; key contributors have been recognised with the 1996, 2007 and 2013 ACM Turing Awards. Here, we restrict our discussion to algorithms that are the basis of the model checkers for SystemVerilog that available to hardware engineers today as well as on the techniques that underpin this work.

Temporal logic describes the intended behaviour of systems and SystemVerilog Assertions—which is based on LTL—is a widely adopted language for this purpose [48, 82]. Any temporal specifications are compositions of safety and liveness properties, where the former indicate the dangerous conditions to be avoided and the latter indicate the desirable conditions to be attained [8, 65]. Safety properties are a fragment of LTL, and can be checked using BDDs by forward fixed-point iterations [12, 34, 62]. Bounded model checking uses SAT and scales much better than BDDs [16], but it is only complete when the bound reaches an often unrealistically large completeness threshold [59]. SAT-based unbounded safety checking uses sophisticated Craig Interpolation and IC3 algorithms [20, 73, 87].

Our work uses a one-step bounded model checking query to check the ranking function (see Eq. (5)), and goes beyond safety. Liveness checking for branching-time CTL is straightforward to implement using BDD-based fixed points [35, 45]. Our method does not support CTL; this is considered acceptable given the prevailing use of LTL-based property languages in industry. LTL model checking is commonly reduced to the fair emptiness problem and, for this purpose, bounded model checking has been generalised to $k$-liveness [31, 56], IC3 has been augmented with strongly connected components [23], and BDD-based algorithms with the Emerson-Lei fixed-point computation [23, 46]. Iterative symbolic computation is the bottleneck on systems with word-level arithmetic. This is usually addressed by either computing succinct explicit-state abstractions of the system [6, 32], or by computing proof certificates based on inductive invariants and ranking functions.

Ranking functions were introduced for termination analysis of software [47], and subsequently generalised to liveness verification [5, 37, 39, 43, 51, 67, 95]. Software and hardware model checking share common questions [42, 76, 77]. Early symbolic approaches for software analysis based on constraint solving are limited to linear ranking functions [21, 84]. As we illustrate in Figure 4, even simple examples often require non-linear ranking functions. These include piecewise-defined functions [63, 93, 94], word-level arithmetic functions [29, 40], lexicographic ranking functions [22, 68], and disjoint well-founded relations [36, 38, 61, 83], and similar proof certificates based on liveness-to-safety translation to reason about the transitive closure of the system [18, 78, 85].

Our method follows a much more lightweight approach than the symbolic approaches above, by training ranking functions from synthetic executions [81]. Deep learning has been successfully applied to generate software and hardware designs, but without delivering any formal guarantees [30, 69, 74]. In our work, we use neural networks *to represent* formal proof certificates, rather than *to generate* proofs or designs. This goes along the lines of recent work on neural certificates, previously applied to control [1, 2, 28, 41, 66, 79, 86, 99–102], formal verification of software and probabilistic programs [3, 4, 50], and this work applies them for the first time to hardware model checking.

# 7 Conclusion

We have introduced a method that leverages (quantised) neural networks as representations of ranking functions for fair termination, which we train from synthetic executions of the system without using any external information other than the design at hand and its specification. We have applied our new method to model checking SystemVerilog Assertions and compared its performance with the state of the art on a range of SystemVerilog designs. We employed off-the-shelf SMT solving (Bitwuzla) and bounded model checking (EBMC) to formally verify our neural ranking functions [76, 80]; although this phase takes the majority of our compute time, with a straightforward implementation and using tiny feed-forward neural networks, we obtained scalability superior to traditional symbolic model checking. Whether alternative neural architectures as well as specialised solvers for quantised neural networks can further improve our approach is topic of future work [11, 64, 72, 75].

This is the first successful application of neural certificates to model checking temporal logic, and introduces hardware model checking as a new application domain for this technology. Neural networks could be used in many other ways to improve model checking. Our work creates a baseline for further development in this field and positively contributes to the safety assurance of systems.

## Acknowledgements

We thank Matthew Leeke, Sonia Marin, and Mark Ryan for their feedback and the anonymous reviewers for their comments and suggestions on this manuscript. This work was supported in part by the Advanced Research + Invention Agency (ARIA) under the Safeguarded AI programme.

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

## A    Details of the SMT Encoding of Quantised Neural Networks

The $k^{\text{th}}$ hidden layer in our network comprises a fully connected layer followed by a clamp operation that restricts outputs to the range $[0, u]$. This layer has $h_k$ neurons, and the previous layer contains $h_{k-1}$ neurons. Each neuron $i$ in the $k^{\text{th}}$ layer is defined by:

$$x_i^{(k)} = \text{Clamp}(y_i^{(k)}; u), \quad y_i^{(k)} = b_i^{(k)} + z_i^{(k)}, \quad z_i^{(k)} = \sum_{j=1}^{h_{k-1}} w_{ij}^{(k-1)} x_j^{(k-1)} \tag{6}$$

To facilitate SMT-checking modulo Bit-Vector theory, we quantise the floating-point weights $w_{ij}$ and biases $b_i$ by multiplying them by $2^f$ and truncating decimals, where $f$ determines the precision. We define:

$$\tilde{w}_{ij}^{(k)} = \text{trunc}(w_{ij}^{(k)} \cdot 2^f), \quad \tilde{b}_i^{(k)} = \text{trunc}(b_i^{(k)} \cdot 2^f)$$

This transformation converts weights from floating-point values in $[0, u]$ to integers in $[0, 2^f u]$. To ensure consistency between bit-vector and floating-point arithmetic, the output of each bit-vector encoded component should be equivalent to multiplying the floating-point output by $2^f$ and truncating the decimals. To achieve this, the SMT constraints on the bit-vectors are formulated as follows:

$$\bigwedge_{i=1}^{h_k} \left( \tilde{x}_i^{(k)} = \text{Clamp}(\tilde{y}_i^{(k)}; 2^f u) \wedge \tilde{y}_i^{(k)} = \tilde{b}_i^{(k)} + \text{ashr}(\tilde{z}_i^{(k)}; f) \wedge \tilde{z}_i^{(k)} = \sum_{j=1}^{h_{k-1}} \tilde{w}_{ij}^{(k-1)} \tilde{x}_j^{(k-1)} \right) \tag{7}$$

Here, $\tilde{w}_{ij}^{(k-1)}$ and $\tilde{x}_j^{(k-1)}$ are integers in $[0, 2^f u]$, thus their product remains within $[0, 2^{2f} u^2]$. The sum $\tilde{z}_i^{(k)}$ aggregates $h_k$ such products, resulting in $[0, 2^{2f} u^2 h_k]$. An arithmetic right shift by $f$ bits scales $\tilde{z}_i^{(k)}$ to $[0, 2^f u^2 h_k]$ to align with $\tilde{b}_i$ in $[0, 2^f u]$ (in floating-point arithmetic, the addition would involve values in $[0, u^2 h_k]$ and $[0, u]$). The clamp operation then restricts $\tilde{y}_i^{(k)}$ to $[0, 2^f u]$, ensuring consistency with the floating-point arithmetic, where the value would lie within $[0, u]$.

To prevent overflow in the SMT query, we set bit-vector sizes appropriately. Let $B$ be such that $2^B \geq 2^f u$. Each product $\tilde{w}_{ij}^{(k)} \tilde{x}_j^{(k)}$ requires up to $2B$ bits, and summing $h_k$ terms necessitates additional $\log h_k$ bits.

This encoding is standard in post-training quantisation of fully connected layers [49]. For element-wise multiplication layers, where each input is multiplied by a corresponding weight, we quantise $w_i \cdot x_i$ as $\text{ashr}(\tilde{w}_i \cdot \tilde{x}_i; f)$: Again, $\tilde{w}_i \tilde{x}_i$ lies within $[0, 2^{2f} u^2]$, and the right shift scales it back to $[0, 2^f u^2]$, ensuring consistency with the floating point encoding.

To address the significant slowdown caused by negative numbers in the Bitwuzla SMT-solver during our experiments, we restructured the dot product computation in equation 7. By decomposing the weight vector $\tilde{w}_{ij}$ into two non-negative components—$\tilde{w}_{ij}^+$ containing positive weights and $\tilde{w}_{ij}^-$ containing the absolute values of negative weights—we expressed the linear layers as

$$\sum_{j=1}^{h} \tilde{w}_{ij} \tilde{x}_j = \sum_{j=1}^{h} \tilde{x}_j \tilde{w}_{ij}^+ - \sum_{j=1}^{h} \tilde{x}_j \tilde{w}_{ij}^- \tag{8}$$

This transformation simplified multiplications to involve only non-negative numbers and consolidated negative operations into a single subtraction, speeding up the SMT-check in our experiments.

We further rewrite the SMT encoding—originally involving several $\tilde{a} \cdot \tilde{x}$ multiplications, where $\tilde{x}$ is a neuron value and $\tilde{a}$ is a quantised integer weight—by replacing these multiplications with additions and left shifts. By factorising $\tilde{a}$ as a sum of powers of two, $\tilde{a} = \sum_{i=0}^{d} c_i \cdot 2^i$, where $c_i \in \{0, 1\}$, the multiplication can be rewritten as:

$$\tilde{a} \cdot \tilde{x} = \sum_{i=0}^{d} c_i \cdot \text{shl}(\tilde{x}; i),$$

where $\text{shl}(\tilde{x}; i)$ represents left-shifting $x$ by $i$ bits, effectively multiplying $x$ by $2^i$.

# B   Details of the Case Studies

We consider ten hardware designs in our study. These serve as benchmarks to demonstrate the scalability of our method compared to conventional symbolic model checkers. They are designed to be parameterizable.

The `DELAY` models generates a positive signal `sig` after a fixed delay determined by the counter `cnt`, includes a reset input event that sets `cnt` to $0$, and aims to ensure that `sig` occurs infinitely often under the assumption that the reset event `rst` is received finitely many times, resulting in the specification $\mathsf{FG}$ `!rst` $\rightarrow \mathsf{GF}$ `sig`. We further verify $\mathsf{FG}$ `!rst` $\rightarrow \mathsf{GF}$ (`sig` $\wedge \mathsf{X}$ `!sig`), to ensure `sig` doesn't remain triggered forever.

The `LCD Controller` (LCD) performs a display initialisation setup, then awaits the `lcd_enable` signal to transition from `ready` to `send` for data transmission, and returns to `ready` after a fixed interval, ensuring $\mathsf{FG}$ `lcd_enable` $\rightarrow \mathsf{GF}$ `ready`.

Similarly, `Thermocouple` (Tmcp.) transitions through stages, `start`, `get_data` and `pause` with suitable delay in between, processing SPI transactions and managing transitions based on bus activity, adhering to the specification $\mathsf{FG}$ `!rst` $\rightarrow \mathsf{GF}$ `get_data`.

The `7-Segment` (7-Seg) model alternates between two displays, ensuring each is activated regularly unless reset, as specified by $\mathsf{FG}$ `!rst` $\rightarrow (\mathsf{GF}$ `disp = 0` $\wedge \mathsf{GF}$ `disp = 1`), we also verify a simpler specification $\mathsf{FG}$ `!rst` $\rightarrow \mathsf{GF}$ `disp = 1`.

The `i2c Stretch` (i2cS) generates timing signals `scl_clk` and `data_clk` based on the ratio of input and bus clock frequencies [90, 92]. It monitors `rst` and detects the `ena` signal to manage clock stretching, ensuring $\mathsf{FG}$ (`!rst` $\&$ `ena`) $\rightarrow \mathsf{GF}$ `stretch`.

The `Pulse Width Modulation` (PWM) system utilises an $N$-bit counter to adjust pulse widths dynamically based on input, verifying the low setting of pulse infinitely often as $\mathsf{GF}$ `!pulse` [54].

The `VGA Controller` (VGA) manages a display interface using horizontal and vertical counters for pixel coordinates, ensuring smooth rendering by adjusting sync pulses and the display enable signal `disp_ena`, here we confirm $\mathsf{FG}$ `!rst` $\rightarrow \mathsf{GF}$ `disp_ena`.

The `UART Transmitter` (UARTt) toggles between `wait` for preparing data and `transmit` for sending data, based on `tx_ena` requests and `clk` signals, validated by $\mathsf{FG}$ `!rst` $\rightarrow \mathsf{GF}$ `wait` [90].

The `Load-Store` (LS) toggles between `load` and `store` with a delay implemented by counter which counts from 0 up to N when `load` then switch to `store` counting back down to 0, before switching back to `load`, `sig` signals a switch from `load` to `store`, and we verify $\mathsf{FG}$ `!rst` $\rightarrow \mathsf{GF}$ `sig`.

Lastly, the `Gray Counter` (Gray) counts in Gray codes to minimise transition errors by ensuring single bit changes between consecutive counts, with $\mathsf{FG}$ `!rst` $\rightarrow \mathsf{GF}$ `sig`, indicating regular signalling of complete cycles [97]. Similar to the Delay module, we aim to ensure that the signal sig does not remain triggered indefinitely. We establish this with two distinct specifications $\mathsf{FG}$`!rst` $\rightarrow \mathsf{GF}$(`sig` $\wedge \mathsf{X}$ `!sig`) and $\mathsf{FG}$`!rst` $\rightarrow (\mathsf{GF}$`sig` $\wedge \mathsf{GF}$ `!sig`).

# C   Details of the Experimental Results

Table 3 provides the runtimes for each tool on the 194 verification tasks considered in Section 5. These tasks involve verifying each hardware design across an increasing state space, labelled numerically. The "Train Time" column indicates the training duration for the neural network in seconds, while the other columns represent the total runtime for each tool, with the fastest tool time in bold and the rest in grey. In this table, our method uses the configuration described in Section 5, with two hidden layers containing 8 and 5 neurons, respectively. Some of our runtimes are marked with an asterisk (*), indicating that in those cases we obtained counterexamples using the SMT solver; these were used for retraining and then validating the trained network. The reported time includes all SMT checks and training. Table 1 summaries these results by showing the number of tasks successfully completed by each tool for each design. Tasks not marked as *out of time (oot.)* or *did not train (dnt.)* are considered successful. Table 3 serves as the basis for computing all statistical observations discussed in Section 1 and Section 5, except those related to the "our best" line in Figure 6a. All other

| Model | LTL Specification | Key Table 3 |
|---|---|---|
| DELAY | FG !rst $\rightarrow$ GF sig | Da |
| | FG !rst $\rightarrow$ GF (sig $\wedge$ X !sig) | Db |
| LCD Controller | FG lcd_enable $\rightarrow$ GF ready | L |
| Thermocouple | FG !rst $\rightarrow$ GF get_data | T |
| 7-Segment | FG !rst $\rightarrow$ GF disp = 1 | 7a |
| | FG !rst $\rightarrow$ (GF disp = 0 $\wedge$ GF disp = 1) | 7b |
| i2c Stretch | FG (!rst $\&$ ena) $\rightarrow$ GF stretch | I |
| Pulse Width Modulation | GF !pulse | P |
| VGA Controller | FG !rst $\rightarrow$ GF disp_ena | V |
| UART Transmitter | FG !rst $\rightarrow$ GF wait | U |
| Load-Store | FG !rst $\rightarrow$ GF sig | Ls |
| Gray Counter | FG !rst $\rightarrow$ GF sig | Ga |
| | FG!rst $\rightarrow$ GF(sig $\wedge$ X !sig) | Gb |
| | FG!rst $\rightarrow$ (GFsig $\wedge$ GF !sig) | Gc |

Table 2: Model Name and LTL Specification in our Benchmark

components of Figure 6 are derived from this table. By aggregating the duration of each experiment in the table, including OOT instances counted as 5 hours per experiment, the total time amounts to 104 days and 11 hours.

# D   Ablation Study

The network architecture described in Section 3 includes an element-wise multiplication layer and separate trainable parameters associated with each state of the automaton $\mathcal{A}_{\neg\Phi}$. For most of our experiments in Section 5 and all experiments in Appendix C, we employ a fully connected multilayer perceptron component with two hidden layers containing 8 and 5 neurons, respectively. To experimentally justify our architecture, we perform an ablation study and report the runtimes for different configurations in Table 4. We consider three configurations for the two hidden layers: containing (3, 2) neurons, (5, 3) neurons, and (15, 8) neurons, respectively. We further replace the element-wise multiplication layer with a fully connected layer of the same size, denoted as 'ExtL' for the extra layer. Additionally, we explore providing the global trainable parameters $\theta$ to all automaton states of the automaton $\mathcal{A}_{\neg\Phi}$, leading to a monolithic neural ranking function $V(r,q) \equiv \bar{V}(r,q;\theta)$, where the automaton state $q$ is given as an additional input, which we denote as 'Mono'.

Given the large number of possible combinations of these modifications, we restrict our ablation study to switching only a single configuration at a time. In Table 4, the column labelled 'Default' contains the results for our original configuration—the runtimes in this column are the same as those under 'Our (8, 5)' in Table 3. Following that, we have one column for each of the three hidden layer configurations, followed by columns for the extra layer ('ExtL'), and the monolithic neural ranking function ('Mono'). The 'our best' line in Figure 6a is obtained by selecting the minimum runtime from the 'Default' and the three hidden layer configuration columns for each of the 194 tasks.

From Table 4, we observe that our default configuration succeeds in more cases than the alternative configurations, justifying our choices experimentally. Specifically, the default configuration completes 93 % of the tasks, while the three configurations with hidden layers containing (3, 2), (5, 3), and (15, 8) neurons complete 25 %, 63 %, and 74 % of the tasks, respectively. The extra-layer configuration and the monolithic neural ranking function complete 24 %, and 39 %, of the tasks, respectively.

Generally—but not always—when a smaller network succeeds, its runtime is lower than that of the default network. Specifically, among the tasks completed by the (3, 2) neuron configuration, it

was faster than the default configuration in $57\%$ of cases; for the (5, 3) neuron configuration, this statistic rises to 94%. Interestingly, this trend does not hold when comparing the (3, 2) and (5, 3) configurations: despite having more neurons, the (5, 3) configuration was faster than the (3, 2) configuration in $56\%$ of tasks. The default configuration not only completes more tasks than the (15, 8) configuration but is also faster on $97\%$ of the tasks successfully completed by the (15, 8) configuration. Notably, among the hidden layer configurations only the (15, 8) configuration succeeds on any of the tasks for the `VGA` design, labelled as 'V' in the table. In $67\%$ of the tasks that the 'Ext. L' configuration completes, it is faster than the default configuration; this figure rises to $86\%$ for the 'Mono' configuration. While the monolithic neural ranking function ('Mono') fails on 61% of tasks, it surprisingly succeeds on nine out of the ten tasks for the `VGA` design. Overall, only 5 of the 194 tasks fail under all configurations in the ablation study.


Table 3: Runtime comparison with the state of the art on individual tasks.

| Tasks | Train Time | our (8,5) | nuXmv | ABC | X | Y |
|---|---|---|---|---|---|---|
| $Da_1$ | 6 | 44 | **2.5** | 398 | 442 | oot. |
| $Da_2$ | 10 | 51 | **7** | 1759 | 802 | oot. |
| $Da_3$ | 7 | 80 | **29** | 8666 | 801 | oot. |
| $Da_4$ | 7 | 92 | **121** | oot. | 815 | oot. |
| $Da_5$ | 41 | **157** | 292 | oot. | 788 | oot. |
| $Da_6$ | 24 | **162** | 529 | oot. | 814 | oot. |
| $Da_7$ | 15 | **197** | 870 | oot. | 809 | oot. |
| $Da_8$ | 36 | **214** | 1277 | oot. | 793 | oot. |
| $Da_9$ | 23 | **321** | 1809 | oot. | 809 | oot. |
| $Da_{10}$ | 15 | **306** | 2448 | oot. | 804 | oot. |
| $Da_{11}$ | 44 | **390** | 3516 | oot. | 789 | oot. |
| $Da_{12}$ | 17 | **412** | 4461 | oot. | 788 | oot. |
| $Da_{13}$ | 22 | **674** | oot. | oot. | 808 | oot. |
| $Da_{14}$ | 26 | 1500 | oot. | oot. | **815** | oot. |
| $Da_{15}$ | 45 | 3365 | oot. | oot. | **802** | oot. |
| $Da_{16}$ | 124 | 8684 | oot. | oot. | **813** | oot. |
| $P_1$ | 2 | 30 | **1.5** | 926 | 792 | oot. |
| $P_2$ | 1 | 26 | **6.5** | 5087 | 776 | oot. |
| $P_3$ | 4 | 97 | **30** | oot. | 785 | oot. |
| $P_4$ | 12 | **69** | 137 | oot. | 782 | oot. |
| $P_5$ | 14 | **94** | 638 | oot. | 788 | oot. |
| $P_6$ | 30 | **177** | 2563 | oot. | 773 | oot. |
| $P_7$ | 64 | **365** | 10667 | oot. | 787 | oot. |
| $P_8$ | 334 | 1028 | oot. | oot. | **798** | oot. |
| $P_9$ | 17 | 2611 | oot. | oot. | **781** | oot. |
| $P_{10}$ | 20 | 6527 | oot. | oot. | **788** | oot. |
| $P_{11}$ | 33 | 9353 | oot. | oot. | **787** | oot. |
| $P_{12}$ | dnt. | - | oot. | oot. | **785** | oot. |
| $L_1$ | 4 | 42 | **0.8** | 129 | 55 | oot. |
| $L_2$ | 5 | 63 | **1.9** | 1189 | 215 | oot. |
| $L_3$ | 5 | 53 | **12** | 1712 | 808 | oot. |
| $L_4$ | 8 | 83 | **12** | oot. | 799 | oot. |
| $L_5$ | 46 | **245** | 951 | oot. | 838 | oot. |
| $L_6$ | 18 | **297** | 4444 | oot. | 815 | oot. |
| $L_7$ | 22 | **360** | 3262 | oot. | 843 | oot. |
| $L_8$ | 31 | **335** | 11061 | oot. | 828 | oot. |
| $L_9$ | 38 | **355** | 1743 | oot. | 807 | oot. |
| $L_{10}$ | 32 | **470** | oot. | oot. | 837 | oot. |
| $L_{11}$ | 23 | **360** | oot. | oot. | 825 | oot. |
| $L_{12}$ | 102 | **622** | oot. | oot. | 805 | oot. |
| $L_{13}$ | 79 | 3321 | oot. | oot. | **850** | oot. |
| $L_{14}$ | 181 | 7133 | oot. | oot. | **817** | oot. |
| $I_1$ | 6 | 49 | **2.5** | 201 | 169 | oot. |
| $I_2$ | 9 | 74 | **10** | 1195 | oot. | oot. |
| $I_3$ | 12 | 102 | **44** | 6396 | 793 | oot. |
| $I_4$ | 78 | **163** | 196 | oot. | 801 | oot. |
| $I_5$ | 42 | **170** | 527 | oot. | 795 | oot. |
| $I_6$ | 25 | **173** | 1038 | oot. | 797 | oot. |
| $I_7$ | 29 | **173** | 1801 | oot. | 794 | oot. |
| $I_8$ | 62 | **245** | 2738 | oot. | 800 | oot. |
| $I_9$ | 49 | **289** | 9288 | oot. | 797 | oot. |
| $I_{10}$ | 63 | **474** | 15674 | oot. | oot. | oot. |
| $I_{11}$ | 62 | **354** | oot. | oot. | 807 | oot. |
| $I_{12}$ | 97 | **386** | oot. | oot. | 805 | oot. |
| $I_{13}$ | 134 | **358** | oot. | oot. | 798 | oot. |
| $I_{14}$ | 114 | **417** | oot. | oot. | 818 | oot. |
| $I_{15}$ | 342 | **661** | oot. | oot. | 792 | oot. |
| $I_{16}$ | 140 | **585.** | oot. | oot. | 812 | oot. |
| $I_{17}$ | 332 | **615.** | oot. | oot. | 798 | oot. |
| $I_{18}$ | 474 | **908.** | oot. | oot. | 824 | oot. |
| $I_{19}$ | oot. | oot.. | oot. | oot. | **817** | oot. |
| $I_{20}$ | oot. | oot.. | oot. | oot. | **811** | oot. |
| $Ga_1$ | 3 | 18 | **0.3** | 53 | 81 | oot. |
| $Ga_2$ | 3 | 30 | **1.2** | 78 | 293 | oot. |
| $Ga_3$ | 6 | 25 | **5** | 233 | 784 | oot. |
| $Ga_4$ | 10 | 37 | **22** | 6490 | 795 | oot. |
| $Ga_5$ | 4 | **62** | 96 | 6217 | 789 | oot. |
| $Ga_6$ | 12 | **102** | 447 | oot. | 786 | oot. |
| $Ga_7$ | 17 | **175** | 2062 | oot. | 801 | oot. |
| $Ga_8$ | 28 | **299** | 12935 | oot. | 797 | oot. |
| $Ga_9$ | 39 | **639** | oot. | oot. | 811 | oot. |
| $Ga_{10}$ | 118 | 1566 | oot. | oot. | **792** | oot. |
| $Ga_{11}$ | 218 | 5790 | oot. | oot. | **787** | oot. |
| $Db_1$ | 12 | 89 | **3** | 231 | 993 | oot. |
| $Db_2$ | 10 | 85 | **7** | 379 | 6568 | oot. |
| $Db_3$ | 12 | 181 | **30** | 3396 | oot. | oot. |
| $Db_4$ | 14 | 204 | **130** | oot. | oot. | oot. |
| $Db_5$ | 30 | **312** | 308 | oot. | 4931 | oot. |
| $Db_6$ | 145 | **471** | 570 | oot. | oot. | oot. |
| $Db_7$ | 158 | **711** | 917 | oot. | oot. | oot. |
| $Db_8$ | 170 | **532** | 1349 | oot. | oot. | oot. |
| $Db_9$ | 25 | **662** | 1912 | oot. | oot. | oot. |
| $Db_{10}$ | 214 | **746** | 2605 | oot. | oot. | oot. |
| $Db_{11}$ | 226 | **885** | 3597 | oot. | oot. | oot. |
| $Db_{12}$ | 200 | **930** | 4439 | oot. | oot. | oot. |
| $Db_{13}$ | 363 | **2654** | oot. | oot. | oot. | oot. |
| $Db_{14}$ | 728 | **3893** | oot. | oot. | oot. | oot. |
| $Db_{15}$ | 588 | **5700** | oot. | oot. | oot. | oot. |
| $Db_{15}$ | 797 | **12697** | oot. | oot. | oot. | oot. |
| $Ls_1$ | 6 | 51 | **16** | 768 | 510 | oot. |
| $Ls_2$ | 5 | **53** | 56 | 10772 | 539 | oot. |
| $Ls_3$ | 3 | **78** | 251 | oot. | 580 | oot. |
| $Ls_4$ | 19 | **126** | 1263 | oot. | 621 | oot. |
| $Ls_5$ | 21 | **185** | 2612 | oot. | 633 | oot. |
| $Ls_6$ | 26 | **218** | 6722 | oot. | 662 | oot. |
| $Ls_7$ | 22 | **403** | 9490 | oot. | 668 | oot. |
| $Ls_8$ | 24 | **300** | 12665 | oot. | 674 | oot. |

| Tasks | Train Time | our (8,5) | nuXmv | ABC | X | Y |
|---|---|---|---|---|---|---|
| $7a_1$ | 5 | 21 | **2** | 39 | 28 | oot. |
| $7a_2$ | 5 | 34 | **8** | 119 | 192 | oot. |
| $7a_3$ | 4 | **24** | 70 | 614 | 467 | oot. |
| $7a_4$ | 5 | **38** | 1405 | 1469 | 680 | oot. |
| $7a_5$ | 5 | **47** | 11605 | oot. | 812 | oot. |
| $7a_6$ | 4 | **58** | oot. | oot. | 806 | oot. |
| $7a_7$ | 5 | **86** | oot. | oot. | 820 | oot. |
| $7a_8$ | 6 | **104** | oot. | oot. | 815 | oot. |
| $7a_9$ | 15 | **215** | oot. | oot. | 816 | oot. |
| $7a_{10}$ | 10 | **154** | oot. | oot. | 816 | oot. |
| $7a_{11}$ | 11 | **242** | oot. | oot. | 817 | oot. |
| $7a_{12}$ | 20 | **208** | oot. | oot. | 817 | oot. |
| $7a_{13}$ | 18 | **495** | oot. | oot. | 815 | oot. |
| $7a_{14}$ | 26 | 977 | oot. | oot. | **819** | oot. |
| $7a_{15}$ | 68 | 2077 | oot. | oot. | **824** | oot. |
| $T_1$ | 7 | 22 | **0.6** | 2 | 1 | oot. |
| $T_2$ | 23 | 67 | **9** | 62 | 470 | oot. |
| $T_3$ | 11 | **95** | 361 | 234 | 41 | oot. |
| $T_4$ | 11 | **98** | 601 | 344 | 103 | oot. |
| $T_5$ | 16 | **164** | 306 | 1872 | 414 | oot. |
| $T_6$ | 12 | **156** | 573 | 1246 | 246 | oot. |
| $T_7$ | 12 | **182** | 1192 | 2195 | 308 | oot. |
| $T_8$ | 23 | **210** | 1935 | oot. | 532 | oot. |
| $T_9$ | 17 | **326** | 4224 | oot. | 798 | oot. |
| $T_{10}$ | 20 | **516** | 6365 | oot. | 796 | oot. |
| $T_{11}$ | 44 | **389** | 9691 | oot. | 797 | oot. |
| $T_{12}$ | 39 | 932 | 15129 | oot. | **791** | oot. |
| $T_{13}$ | 39 | 949 | oot. | oot. | **807** | oot. |
| $T_{14}$ | 104 | 1552 | oot. | oot. | **803** | oot. |
| $T_{15}$ | 79 | 6020 | oot. | oot. | **805** | oot. |
| $T_{16}$ | 60 | 7331 | oot. | oot. | **800** | oot. |
| $T_{17}$ | 118 | 13042 | oot. | oot. | **786** | oot. |
| $V_1$ | dnt. | - | **25** | 26 | 90 | oot. |
| $V_2$ | dnt. | - | **781** | oot. | 794 | oot. |
| $V_3$ | dnt. | - | 4448 | 2870 | **796** | oot. |
| $V_4$ | dnt. | - | oot. | 4748 | **795** | oot. |
| $V_5$ | dnt. | - | oot. | oot. | **792** | oot. |
| $V_6$ | dnt. | - | oot. | oot. | **792** | oot. |
| $V_7$ | dnt. | - | oot. | oot. | **793** | oot. |
| $V_8$ | dnt. | - | oot. | oot. | **801** | oot. |
| $V_9$ | dnt. | - | oot. | oot. | **805** | oot. |
| $V_{10}$ | dnt. | - | oot. | oot. | **841** | oot. |
| $U_1$ | 7 | 35 | **0.04** | 0.48 | 1.34 | oot. |
| $U_2$ | 11 | 29 | **0.06** | 0.39 | 1.3 | oot. |
| $U_3$ | 19 | 148 | **0.08** | 0.45 | 1.94 | oot. |
| $U_4$ | 53 | 74 | **0.4** | 0.45 | 1.21 | oot. |
| $U_5$ | 103 | 188 | **0.24** | 0.42 | 1.22 | oot. |
| $U_6$ | 295 | 371 | **0.09** | 0.39 | 1.25 | oot. |
| $U_7$ | 26 | 222 | **0.14** | 0.49 | 1.32 | oot. |
| $U_8$ | 514. | 1804 | **0.1** | 0.46 | 1.21 | oot. |
| $U_9$ | 51 | 567 | **0.12** | 0.72 | 1.29 | oot. |
| $U_{10}$ | 46 | 112 | 3.28 | **0.79** | 1.41 | oot. |
| $Gb_1$ | 3 | 41 | **0.3** | 49 | 824 | oot. |
| $Gb_2$ | 3 | 97 | **1** | 196 | 417 | oot. |
| $Gb_3$ | 3 | 160 | **5** | 358 | 3204 | oot. |
| $Gb_4$ | 3 | 207 | **24** | 1559 | 3661 | oot. |
| $Gb_5$ | 5 | 302 | **110** | oot. | 13164 | oot.. |
| $Gb_6$ | 9 | **292** | 511 | oot. | oot. | oot. |
| $Gb_7$ | 7 | **862** | 2441 | oot. | 3341 | oot. |
| $Gb_8$ | 8 | **2958** | 14518 | oot. | oot. | oot. |
| $Gb_9$ | 10 | **3847** | oot. | oot. | oot. | oot. |
| $Gb_{10}$ | 18 | **4676** | oot. | oot. | oot. | oot. |
| $Gb_{11}$ | 36 | **8834** | oot. | oot. | oot. | oot. |
| $Gc_1$ | 8 | 41 | **0.3** | 88 | 94 | oot. |
| $Gc_2$ | 12 | 52 | **1.25** | 139 | 539 | oot. |
| $Gc_3$ | 5 | 100 | **5** | 4428 | 3349 | oot. |
| $Gc_4$ | 6 | 132 | **24** | 4373 | 3688 | oot. |
| $Gc_5$ | 73 | 260 | **105** | oot. | oot. | oot. |
| $Gc_6$ | 17 | **256** | 491 | oot. | 3488 | oot. |
| $Gc_7$ | 50 | **1091** | 2387 | oot. | oot. | oot. |
| $Gc_8$ | 176 | **947** | 14287 | oot. | oot. | oot. |
| $Gc_9$ | 580 | **2300** | oot. | oot. | oot. | oot. |
| $Gc_{10}$ | 1685 | **5052** | oot. | oot. | oot. | oot. |
| $Gc_{11}$ | 169 | **9888** | oot. | oot. | oot. | oot. |
| $7b_1$ | 5 | 30 | **1.5** | 69 | 558 | oot. |
| $7b_2$ | 6 | 60 | **5** | 350 | 5352 | oot. |
| $7b_3$ | 5 | 59 | **20** | 3606 | oot. | oot. |
| $7b_4$ | 5 | **75** | 1463 | 2663 | 2332 | oot. |
| $7b_5$ | 5 | **102** | 13208 | oot. | oot. | oot. |
| $7b_6$ | 6 | **125** | oot. | oot. | oot. | oot. |
| $7b_7$ | 7 | **181** | oot. | oot. | oot. | oot. |
| $7b_8$ | 12 | **207** | oot. | oot. | oot. | oot. |
| $7b_9$ | 9 | **438** | oot. | oot. | oot. | oot. |
| $7b_{10}$ | 14 | **238** | oot. | oot. | oot. | oot. |
| $7b_{11}$ | 15 | **439*** | oot. | oot. | oot. | oot. |
| $7b_{12}$ | 14 | **343** | oot. | oot. | oot. | oot. |
| $7b_{13}$ | 22 | **578** | oot. | oot. | oot. | oot. |
| $7b_{14}$ | 65 | **2121*** | oot. | oot. | oot. | oot. |
| $7b_{15}$ | 48 | **2187** | oot. | oot. | oot. | oot. |
| $Ls_9$ | 24 | **473** | oot. | oot. | 709 | oot. |
| $Ls_{10}$ | 19 | **486** | oot. | oot. | 703 | oot. |
| $Ls_{11}$ | 22 | **558** | oot. | oot. | 727 | oot. |
| $Ls_{12}$ | 75 | **695** | oot. | oot. | 709 | oot. |
| $Ls_{13}$ | 22 | 1420 | oot. | oot. | **750** | oot. |
| $Ls_{14}$ | 125 | 4336 | oot. | oot. | **791** | oot. |
| $Ls_{15}$ | 197 | 14533* | oot. | oot. | **832** | oot. |
| $Ls_{15}$ | 88 | oot.* | oot. | oot. | **873** | oot. |

Table 4: Ablation Study Runtime.

| Tasks | Total Time per Setup (in sec.) | | | | | |
|---|---|---|---|---|---|---|
| | Default | (3, 2) | (5, 3) | (15, 8) | Ext.L | Mono |
| $Da_1$ | 44 | 14 | 21 | 121 | 29 | 44 |
| $Da_2$ | 51 | 11 | 23 | 172 | 30 | 66 |
| $Da_3$ | 80 | 15 | 27 | 324 | 64 | 48 |
| $Da_4$ | 92 | 21 | 37 | 330 | 71 | 53 |
| $Da_5$ | 157 | 47 | 41 | 609 | 81 | 57 |
| $Da_6$ | 162 | 90 | 52 | 410 | 84 | fail |
| $Da_7$ | 197 | 169 | 119 | 427 | 84 | 64 |
| $Da_8$ | 214 | 182 | 109 | 595 | 71 | fail |
| $Da_9$ | 321 | 221 | 104 | 789 | 115 | 160 |
| $Da_{10}$ | 306 | 255 | 132 | 654 | 125 | fail |
| $Da_{11}$ | 390 | 420 | 161 | 1093 | 192 | fail |
| $Da_{12}$ | 412 | 383 | 177 | 1200 | 135 | fail |
| $Da_{13}$ | 674 | 443 | 204 | 1446 | fail | 375 |
| $Da_{14}$ | 1500 | 686 | 618 | 3463 | fail | 643 |
| $Da_{15}$ | 3365 | 1239 | 2217 | 6525 | fail | fail |
| $Da_{16}$ | 8684 | 4526 | 3272 | fail | fail | 1597 |
| $P_1$ | 30 | 65 | 15 | 83 | 18 | 18 |
| $P_2$ | 26 | 77 | 12 | 109 | 16 | 85 |
| $P_3$ | 97 | 72 | 14 | 255 | 44 | 25 |
| $P_4$ | 69 | 85 | 20 | fail | 89 | 71 |
| $P_5$ | 94 | 601 | 24 | fail | 102 | 41 |
| $P_6$ | 177 | 374 | 68 | fail | 147 | 119 |
| $P_7$ | 365 | 695 | 154 | fail | 241 | 248 |
| $P_8$ | 1028 | 1364 | fail | fail | 462 | 355 |
| $P_9$ | 2611 | 6030 | fail | fail | 1561 | fail |
| $P_{10}$ | 6527 | 5667 | fail | fail | 1597 | 1943 |
| $P_{11}$ | 9353 | 29992 | fail | fail | 8019 | 8190 |
| $P_{12}$ | fail | fail | fail | fail | fail | fail |
| $L_1$ | 42 | fail | 15 | 172 | fail | 30 |
| $L_2$ | 63 | fail | 23 | 349 | fail | 29 |
| $L_3$ | 53 | fail | 34 | 335 | fail | fail |
| $L_4$ | 83 | fail | 32 | 575 | fail | fail |
| $L_5$ | 245 | fail | 51 | 1043 | fail | 173 |
| $L_6$ | 297 | fail | 116 | 829 | fail | fail |
| $L_7$ | 360 | fail | 234 | 1106 | fail | fail |
| $L_8$ | 335 | fail | fail | 1329 | fail | fail |
| $L_9$ | 355 | fail | fail | 1328 | fail | fail |
| $L_{10}$ | 470 | fail | 297 | 2751 | fail | fail |
| $L_{11}$ | 360 | fail | 429 | 1679 | fail | 179 |
| $L_{12}$ | 622 | fail | 224 | 2660 | fail | fail |
| $L_{13}$ | 3321 | fail | 1905 | 7537 | fail | 929 |
| $L_{14}$ | 7133 | fail | 3041 | fail | fail | fail |
| $I_1$ | 49 | fail | fail | fail | fail | 56 |
| $I_2$ | 74 | fail | fail | 288 | fail | 73 |
| $I_3$ | 102 | fail | fail | fail | fail | 368 |
| $I_4$ | 121 | fail | 921 | fail | fail | fail |
| $I_5$ | 170 | fail | fail | 596 | 254 | fail |
| $I_6$ | 173 | fail | fail | fail | fail | fail |
| $I_7$ | 173 | fail | fail | 987 | fail | fail |
| $I_8$ | 245 | fail | fail | fail | fail | fail |
| $I_9$ | 289 | fail | fail | 815 | fail | fail |
| $I_{10}$ | 474 | fail | fail | 1637 | fail | fail |
| $I_{11}$ | 354 | fail | fail | 1595 | fail | fail |
| $I_{12}$ | 386 | fail | fail | 1016 | fail | fail |
| $I_{13}$ | 358 | fail | fail | fail | 754 | fail |
| $I_{14}$ | 417 | fail | fail | 1116 | fail | fail |
| $I_{15}$ | 661 | fail | fail | fail | fail | fail |
| $I_{16}$ | 585 | fail | fail | 1499 | fail | fail |
| $I_{17}$ | 615 | fail | fail | fail | fail | fail |
| $I_{18}$ | 908 | fail | fail | 2372 | fail | fail |
| $I_{19}$ | fail | fail | fail | fail | fail | fail |
| $I_{20}$ | fail | fail | fail | fail | fail | fail |
| $Ga_1$ | 18 | 5 | 9 | 66 | 22 | 8 |
| $Ga_2$ | 30 | 5 | 12 | 91 | 14 | 18 |
| $Ga_3$ | 25 | 9 | 19 | 148 | 30 | 25 |
| $Ga_4$ | 37 | 12 | 27 | 346 | 47 | 15 |
| $Ga_5$ | 62 | 14 | 30 | 425 | 56 | 46 |
| $Ga_6$ | 102 | 39 | 40 | 309 | 73 | 81 |
| $Ga_7$ | 175 | 113 | 124 | 648 | 101 | 170 |
| $Ga_8$ | 299 | 444 | 163 | 1040 | fail | 178 |
| $Ga_9$ | 639 | 648 | 308 | 2178 | fail | fail |
| $Ga_{10}$ | 1566 | 1014 | 849 | 4727 | fail | 1395 |
| $Ga_{11}$ | 5790 | 2207 | 3318 | 10035 | fail | 1480 |
| $Db_1$ | 89 | fail | 27 | fail | fail | fail |
| $Db_2$ | 85 | fail | 24 | 380 | fail | fail |
| $Db_3$ | 181 | fail | 50 | 1066 | fail | fail |
| $Db_4$ | 204 | fail | 79 | 1276 | fail | fail |
| $Db_5$ | 312 | fail | 187 | 2264 | fail | fail |
| $Db_6$ | 471 | fail | fail | 1889 | fail | fail |
| $Db_7$ | 711 | fail | fail | 1971 | fail | fail |
| $Db_8$ | 532 | fail | fail | fail | fail | fail |
| $Db_9$ | 662 | fail | fail | 2706 | fail | fail |
| $Db_{10}$ | 746 | fail | fail | fail | fail | fail |
| $Db_{11}$ | 885 | fail | fail | 2949 | fail | fail |
| $Db_{12}$ | 930 | fail | fail | 3292 | fail | fail |
| $Db_{13}$ | 2654 | fail | fail | fail | fail | fail |
| $Db_{14}$ | 3893 | fail | fail | 6326 | fail | fail |
| $Db_{15}$ | 5700 | fail | fail | fail | fail | fail |
| $Db_{16}$ | 12697 | fail | fail | fail | fail | fail |

| Tasks | Total Time per Setup (in sec.) | | | | | |
|---|---|---|---|---|---|---|
| | Default | (3, 2) | (5, 3) | (15, 8) | Ext.L | Mono |
| $7a_1$ | 21 | 16 | fail | 73 | fail | fail |
| $7a_2$ | 34 | fail | fail | 130 | fail | fail |
| $7a_3$ | 24 | 484 | fail | 160 | fail | fail |
| $7a_4$ | 38 | 675 | fail | 189 | fail | 14 |
| $7a_5$ | 47 | fail | fail | 199 | fail | fail |
| $7a_6$ | 58 | 210 | fail | 349 | fail | fail |
| $7a_7$ | 86 | fail | fail | 329 | fail | fail |
| $7a_8$ | 104 | 432 | fail | 479 | fail | 118 |
| $7a_9$ | 215 | 611 | 64 | 611 | fail | fail |
| $7a_{10}$ | 154 | 542 | fail | 641 | fail | fail |
| $7a_{11}$ | 242 | 574 | fail | 858 | fail | 164 |
| $7a_{12}$ | 208 | 790 | fail | 835 | fail | fail |
| $7a_{13}$ | 495 | 981 | fail | 1109 | fail | 399 |
| $7a_{14}$ | 977 | 988 | fail | 4137 | fail | fail |
| $7a_{15}$ | 2077 | 1579 | fail | 7811 | fail | fail |
| $T_1$ | 22 | fail | 12 | 70 | fail | 16 |
| $T_2$ | 67 | fail | 30 | 221 | fail | fail |
| $T_3$ | 95 | fail | 46 | 235 | fail | 42 |
| $T_4$ | 98 | fail | 45 | 364 | fail | 54 |
| $T_5$ | 164 | fail | 59 | 269 | fail | 96 |
| $T_6$ | 156 | fail | 76 | 316 | fail | 57 |
| $T_7$ | 182 | fail | 94 | 320 | 162 | fail |
| $T_8$ | 210 | fail | 85 | 377 | fail | fail |
| $T_9$ | 326 | fail | 330 | 1088 | fail | 115 |
| $T_{10}$ | 516 | fail | 179 | 1131 | fail | fail |
| $T_{11}$ | 389 | fail | 246 | 3382 | fail | 263 |
| $T_{12}$ | 932 | fail | 522 | 3846 | fail | 451 |
| $T_{13}$ | 949 | fail | 509 | 3931 | fail | fail |
| $T_{14}$ | 1552 | fail | 730 | 4121 | fail | 1023 |
| $T_{15}$ | 6020 | fail | 1613 | fail | fail | 1276 |
| $T_{16}$ | 7331 | fail | 4896 | fail | fail | 10333 |
| $T_{17}$ | 13042 | fail | 8295 | fail | 8406 | fail |
| $V_1$ | fail | fail | fail | 272 | fail | 82 |
| $V_2$ | fail | fail | fail | fail | fail | 303 |
| $V_3$ | fail | fail | fail | fail | fail | 273 |
| $V_4$ | fail | fail | fail | 2292 | fail | 637 |
| $V_5$ | fail | fail | fail | 3927 | fail | 948 |
| $V_6$ | fail | fail | fail | 15612 | fail | 1135 |
| $V_7$ | fail | fail | fail | fail | fail | 2129 |
| $V_8$ | fail | fail | fail | fail | fail | 3247 |
| $V_9$ | fail | fail | fail | fail | fail | 13628 |
| $V_{10}$ | fail | fail | fail | fail | fail | fail |
| $U_1$ | 35 | fail | 20 | fail | 62 | fail |
| $U_2$ | 29 | fail | 16 | 109 | 234 | 21 |
| $U_3$ | 148 | fail | 24 | 90 | 391 | 105 |
| $U_4$ | 69 | fail | 25 | 593 | 429 | 101 |
| $U_5$ | 74 | fail | 32 | 97 | 760 | 81 |
| $U_6$ | 188 | fail | 34 | 281 | 1792 | 206 |
| $U_7$ | 222 | fail | 42 | 178 | 251 | 195 |
| $U_8$ | 1804 | fail | 81 | fail | fail | 4677 |
| $U_9$ | 567 | fail | 303 | fail | fail | fail |
| $U_{10}$ | 112 | fail | 422 | fail | fail | fail |
| $Gb_1$ | 41 | fail | 41 | 118 | fail | fail |
| $Gb_2$ | 97 | fail | 40 | 268 | fail | fail |
| $Gb_3$ | 160 | fail | 37 | 332 | fail | fail |
| $Gb_4$ | 207 | fail | 61 | fail | fail | fail |
| $Gb_5$ | 302 | fail | 82 | 831 | fail | fail |
| $Gb_6$ | 292 | fail | 162 | fail | fail | fail |
| $Gb_7$ | 862 | fail | 245 | fail | fail | fail |
| $Gb_8$ | 2958 | fail | 427 | 3201 | fail | fail |
| $Gb_9$ | 3847 | fail | 691 | 6981 | fail | fail |
| $Gb_{10}$ | 4676 | fail | 1500 | 17888 | fail | fail |
| $Gb_{11}$ | 8834 | fail | 3820 | fail | fail | fail |
| $Gc_1$ | 41 | fail | 23 | 147 | fail | fail |
| $Gc_2$ | 52 | fail | 21 | 156 | fail | fail |
| $Gc_3$ | 100 | fail | 49 | 288 | fail | fail |
| $Gc_4$ | 132 | fail | 38 | 381 | fail | fail |
| $Gc_5$ | 260 | fail | 54 | 1600 | fail | fail |
| $Gc_6$ | 256 | fail | 123 | 1792 | fail | fail |
| $Gc_7$ | 1091 | fail | 228 | 2343 | fail | fail |
| $Gc_8$ | 947 | fail | 549 | 2864 | fail | fail |
| $Gc_9$ | 2300 | fail | 1470 | 6352 | fail | fail |
| $Gc_{10}$ | 5052 | fail | 2155 | fail | fail | fail |
| $Gc_{11}$ | 169 | 9888 | 6288 | fail | fail | fail |
| $7b_1$ | 30 | fail | 22 | 121 | fail | fail |
| $7b_2$ | 60 | fail | 46 | 217 | fail | fail |
| $7b_3$ | 59 | fail | 33 | 280 | fail | fail |
| $7b_4$ | 75 | fail | 39 | 398 | fail | fail |
| $7b_5$ | 102 | fail | 44 | 752 | fail | fail |
| $7b_6$ | 125 | fail | 60 | 1124 | fail | fail |
| $7b_7$ | 181 | fail | 117 | 996 | fail | fail |
| $7b_8$ | 207 | fail | 143 | 1306 | fail | fail |
| $7b_9$ | 438 | fail | 260 | 3085 | fail | fail |
| $7b_{10}$ | 238 | fail | 198 | 1839 | fail | fail |
| $7b_{11}$ | 439 | fail | 210 | 2349 | fail | fail |
| $7b_{12}$ | 343 | fail | 220 | 1907 | fail | fail |
| $7b_{13}$ | 578 | fail | 366 | 3597 | fail | fail |
| $7b_{14}$ | 2121 | fail | 2070 | 5107 | fail | fail |
| $7b_{15}$ | 2187 | fail | fail | fail | fail | fail |

