# OpenReview forum: "Neural Model Checking"
_NeurIPS.cc/2024/Conference — NeurIPS 2024 poster_

### Official Review · Reviewer_GhN5 · 2024-06-23

**Soundness:** 3
**Presentation:** 3
**Contribution:** 3
**Rating:** 5
**Confidence:** 3

**Summary:**

The paper introduces a novel approach to hardware model checking using neural networks as proof certificates for linear temporal logic (LTL) specifications. Traditional model checking in electronic design automation (EDA) relies on symbolic techniques like SAT solvers, which are computationally intensive. In contrast, the proposed method leverages machine learning to generate neural certificates from random system executions, which are then verified symbolically for validity. The neural network acts as a ranking function ensuring compliance with the LTL specification, trained on synthetic data. Experimental results demonstrate the method's scalability and efficiency, outperforming academic tools in completion time across diverse hardware designs in SystemVerilog, and performing competitively with leading commercial tools.

**Strengths:**

- This approach enhance verification in complex hardware designs efficiently through unsupervised machine learning techniques integrated with symbolic reasoning.
- The paper is well organized and well written.

**Weaknesses:**

- The classic algorithm has a guarantee for the verification. I do not know how the use of deep neural network can still have such guarantee.
- If we transform the problem to the SMT solver with same difficulty, it seems the SMT solver will still take a huge amount of time for verification.

**Questions:**

Can the proposed method provide any formal guarantee over the evaluated program?

**Limitations:**

The author discussed the limitation of this work about the heavy reliance on the SMT solver and the extensibility of application to Computational Tree Logic.

---

> ### Author Rebuttal · Authors · 2024-08-07
>
> * Our method is as formally sound as traditional algorithms. We ensure this soundness through an SMT check, which validates that our neural ranking function is valid across the entire state space. Specifically, the SMT query in Eq. (4) represents the negation of the proof rules outlined in Eqs. (1) and (2). If this formula is unsatisfiable, it confirms that our neural ranking function is formally correct. This methodology aligns with established practices in neural certificate-based approaches, including Neural Lyapunov Control, Neural Termination Analysis, and Neural Supermartingales [25,44,2,58,87].
>
> * It is important to note that the complexity of verifying a ranking function via SMT solving, which is co-NP, is notably less than the complexity of LTL model checking, which is PSPACE-hard. Our machine learning approach heuristically fills this gap by training a proof certificate—a ranking function for fair termination—that we formally validate with an SMT solver. Our experimental results demonstrate that training and verifying a ranking function with SMT is more efficient than applying existing standard model-checking algorithms.
>
> * In summary, our method does provide formal guarantees while demonstrating scalability superior to standard model-checking algorithms.

---

> > ### Comment · Reviewer_GhN5 · 2024-08-13
> > **reply**
> >
> > Thanks for the detailed reply. I'm happy to improve my score from 4 to 5.

---

### Official Review · Reviewer_U2Dq · 2024-06-28

**Soundness:** 3
**Presentation:** 3
**Contribution:** 2
**Rating:** 5
**Confidence:** 4

**Summary:**

This paper addresses the problem of hardware model checking with respect to LTL specifications. Although this a well-studied problem, hardware model checking can still suffer from scalability issues. A general automata-theoretic approach to model checking is to check if intersection of the formal languages corresponding to the system and to the complement of the specification is empty. One strategy for conducting this emptiness check is to construct a ranking function that can serve as an emptiness certificate. The main idea of the paper is to use neural networks to represent and learn these ranking functions. The functions are learned using traces of the system composed with the complement of the specification. It is then checked (using an SMT solver) if the learned function satisfies the required constraints of a ranking function. If not, the counterexamples are used to further train the function but if yes, then we have a proof that the system satisfies its spec. The proposed approach is evaluated on 9 hardware designs and with respect to 123 verification tasks. The presented approach outperforms state-of-the-art baselines on these verification tasks.

**Strengths:**

The paper is largely well-written and, for the chosen benchmarks, the results are quite impressive. The use of neural networks to represent ranking functions has been proposed before (citation [44] in the paper), but the setting in the prior work in slightly different  (termination analysis of programs in [44] vs model checking of hardware ciruits with respect to LTL specifications here). It is interesting to see that neural ranking functions can be effective in the setting of hardware model checking as well. It is particularly striking that the learned ranking function is frequently correct in the first attempt and does not need any further training.

**Weaknesses:**

I have a number of concerns with the paper.
1. Given the prior work on neural termination analysis, I find the technical contribution to be quite incremental. The core of the technical approach is almost the same as in [44].

2. While the empirical results are quite impressive, I wonder how the benchmarks were chosen. Why not choose the benchmarks from HWMCC'20? Also, why not compare against the best performing tool from HWMCC'20, i.e., AVR? Why set the time limit to 5 hours in particular? All these points make me a little concerned about the results. When is this approach not applicable? Can it be applied to all hardware model checking problems? Can it be applied to software model checking problems?

4. The paper may not be easily accessible to readers not familiar with formal methods. In particular, there are a number of approaches for model checking with respect to LTL functions and using ranking functions for the purpose is not that common. I could only find two citations [A,B]. The paper should at least include these citations if not more for the ranking function based LTL model checking approach.

[A] Cook, B., Gotsman, A., Podelski, A., Rybalchenko, A., & Vardi, M. Y. (2007). Proving that programs eventually do something good. ACM SIGPLAN Notices, 42(1), 265-276.

[B] Dietsch, D., Heizmann, M., Langenfeld, V., & Podelski, A. (2015). Fairness modulo theory: A new approach to LTL software model checking. In Computer Aided Verification: 27th International Conference, CAV 2015, San Francisco, CA, USA, July 18-24, 2015, Proceedings, Part I 27 (pp. 49-66). Springer International Publishing.

5. I find the title to overclaim a bit. The paper focuses on hardware model checking but the title suggests that any model checking problem is within scope. Moreover, there might be other ways in which neural networks could be used to aid model checking. Is it a claim of the paper that the only way in which neural networks can aid model checking is as ranking functions? I also think that a formal methods conference might be a better fit for the paper.

**Questions:**

In addition to the questions above, I have the following questions and comments:

1. What if system does not meet its specification? In that case, no ranking function would exist. Is the proposed approach able to handle such scenarios and generate counterexamples when the system violates it spec?
2. The paper frequently refers to word-level but it is never clarified what this means.
3. Fig 1 needs more description. It is a little mysterious at present.
4. Line 101: Is the language L_M defined over sequences of observed states of M or over sequences of atomic propositions about observed states of M. If it is the former, then the language inclusion does not make sense.
5. Line 106: "inputs and observables being equal to obs X" --> this is a little hard to follow
6. Line 161: "normalizes these inputs through element-wise multiplication ..." --> please expand a bit on this
7. Equation 4: The indicator function seems to be incorrectly used. Should the term "- 1_F(q)" instead be "- \epsilon. \neg 1_F(q)"
8. Line 226: Since there are only 9 designs and 123 verification tasks, are there multiple specs per design?
9. Line 238: "network is quantized and translated into SystemVerilog" --> please provide some more details about this
10. Figure 4: I am confused by the figure. How is the state space size and logic gate count different across the same design?
11. Line 288: "Of the 11 tasks that were not trained to zero loss, ..." --> Are these tasks considered as timed out? Are they plotted in Fig 5b?
12. Fig 5b: What are the tasks where x is almost 0 but y has large values?
13. Line 306: Why is different hardware used for experiments with the industrial tools? This makes the head-to-head comparison unfair.

**Limitations:**

Yes, the paper includes a discussion about limitations. I do not see any potential negative societal impact of this work..

---

> ### Author Rebuttal · Authors · 2024-08-07
>
> 1 Our work is inspired by neural certificates [25,44,2,58,87], including neural termination analysis. Previous results on neural certificates focus on reachability/termination and avoidance/safety and temporal logic is largely unexplored. We applied neural certificates to LTL model checking and compared them with state-of-the-art model checkers. This is a substantial step forward in the area of verifying properties using neural certificates and introduces hardware model checking as a new application domain for this technology.
>
> 2 Our benchmarks are derived from extremely common hardware design patterns from standard literature. The HWMCC benchmarks are exclusively bit-level designs and largely focus on safety verification; our work targets word-level designs and general LTL liveness/fairness. We compared with nuXmv and ABC as they are the best-performing tools in the liveness category, whereas AVR is restricted to safety analysis. Our 5-hour limit is significantly beyond the 1-hour limit used in the competition. We applied our approach to hardware model checking, where LTL is a natural specification language. Our approach applies to every LTL verification question and can in principle be extended to the verification of concurrent software, which is subject to future work.
>
> 3 We leverage the fair-termination proof rule applied to Buechi acceptance, introduced in seminal work [45, 59, 83]. We refer to "Proving that programs eventually do something good", which influenced our work [A,33], and give an overview of ranking function synthesis (lines 359-362). The paper "Fairness modulo theory: A new approach to LTL software model checking" is an alternative and important related work in software verification, which we will cite in the final version.
>
> 5 We agree that neural networks could be used in many other ways to improve model checking, and will discuss options for this in our conclusion. We have demonstrated that neural ranking functions are extremely effective for LTL model checking, and we instantiated this technology to circuits and SystemVerilog Assertions. Work on neural certificates is being regularly published at machine learning conferences [25,87,69]. This is a novel and emerging application for neural networks and, for this reason, we believe our result is best suited for NeurIPS. Our work offers a formal guarantee of correctness, a feature which is in strong demand in all AI communities.
>
> Q1 As is standard for methods based on proof rules, the presented approach solves the verification question in one direction. To make an analogy, this is similar to methods based on Hoare logic in software verification, where ranking functions and loop invariants prove correctness but do not directly provide counterexamples. To construct counterexamples, it is standard to couple verifiers with falsifiers such as bounded model checkers. Our prototype is built on top of EBMC and, as such, it includes this capability.
>
> Q2 By word-level we mean hardware descriptions based on arithmetic operations over fixed-width integers, as opposed to Boolean operations over bits. This is an important class of designs and a challenging problem for existing tools. We will clarify this important distinction in the final version.
>
> Q3 We will elaborate on Fig. 1 and clarify the meaning of the outputs "fair" and "rnk".
>
> Q4 The languages L_M and L_Phi are defined over sequences of valuations for the observables of M - the first option suggested by the reviewer. The reader familiar with alphabets defined over atomic propositions can assume the observables to be Boolean, and this will cover the second option.
>
> Q5 Line 106 indicates that an automaton can be expressed symbolically, with variables Y where inp Y = obs Y = obs = X, which is then in parallel composition with the system under analysis. We will elaborate and connect this to Fig. 1.
>
> Q6 Data normalization is a standard practice, typically, datasets are normalized before training. In our approach, we calculate the required normalization factors and incorporate them as the initial non-trainable layer. This enables the neural ranking function to process inputs directly without the need for separate normalization.
>
> Q7 Indeed, the signs of V(r,q;\theta) and V(r',q';\theta) in Eq. (3) should be switched. We will fix this in the final version.
>
> Q8 Each of the 9 designs has one spec, and the 123 tasks are generated by adjusting various parameters of the model, as we elaborate in lines 228-231.
>
> Q9 Quantisation converts a standard neural network over floats into a neural network over fixed-width integers. These are easily mappable to circuits in SystemVerilog. We will give an example in the appendix.
>
> Q10 The designs we use as benchmarks are parameterizable, which is a common feature when designing hardware using Verilog RTL. Our benchmark set includes multiple instances of each design, for a range of typical parameter values.
>
> Q11 Yes, indeed, the tasks that are not trained to zero loss are considered timeouts. These are visible in the top timeout line of Fig 5b.
>
> Q12 The tasks were nuXmv/ABS are fast but our method is slow are problems associated with VGA and UART, and are discussed as limitations (cf. Sec 5)
>
> Q13 We used different hardware because our license for the commercial tools is tied to a specific machine with restricted software. We highlight that we ran our prototype and nuXmv and ABC on the same machine, thus all measurements in Fig. 5 are consistent. We agree on the need for a head-to-head comparison between the industrial and academic tools. We are rerunning all the experiments on a machine with hardware identical to the one used for tools X and Y. We will include these results in the final version. Our preliminary results confirm (unsurprisingly) that our relative runtime w.r.t nuXmv and ABC are consistent with previous results, as we are also running them on the same machine. The results in comparison to X have changed by a small margin (+5%/-5%).

---

> > ### Comment · Reviewer_U2Dq · 2024-08-12
> >
> > Thank you for the detailed response! I will keep my score.

---

### Official Review · Reviewer_2gkW · 2024-07-12

**Soundness:** 3
**Presentation:** 3
**Contribution:** 4
**Rating:** 7
**Confidence:** 4

**Summary:**

The paper presents a novel application of machine learning to hardware model checking. The model checking problem is given as a design written in SystemVerilog and a temporal logic property in linear-time temporal logic (LTL). Similar to many classical model checking approaches the authors first construct the synchronous composition of the system model and the negation of the LTL property. The model checking problem then corresponds to checking whether the composition is empty (under some fairness condition). This emptiness can be witnessed by a ranking function on the states of the synchronous composition. The authors propose to learn this ranking function by representing it as a neural network and training the neural network with trajectories sampled from the composition. After training, a symbolic ranking function is extracted from the neural network using post-training quantization and checked with an SMT solver to be valid. On a set of benchmarks from the literature the authors impressively demonstrate that their approach often outperforms state-of-the-art academic model checkers and is competitive with industrial-grade model checkers.

**Strengths:**

- The model checking problem is a fundamental problem in formal verification and is of great importance in industry. The paper contributes a novel neuro-symbolic approach to this fundamental problem. To the best of my knowledge, it is the first successful application of neural networks to the LTL model checking problem.
- With designs being expressed in a hardware description language and properties being expressed in LTL, the approach is evaluated similarly to a real-world workflow.
- The prototype implementation is already competitive with established tools. The authors compare with nuXmv and ABC which are state-of-the-art solvers in academia and the result of many years of engineering and research. The prototype outperforms those tools on 7 out of 9 benchmarks. Impressively, the authors show that their prototype is also competitive with industrial-grade solvers.
- The results are also interesting for its use of ranking functions in the hardware model checking problem.

**Weaknesses:**

- When being familiar with the LTL model checking problem, Section 2 of the paper gives a helpful description of the problem. However, I expect it to be hard to follow for someone less familiar with the problem.
- The approach requires some tuning when being applied to a specific problem instance (with respect to trajectory sampling and the neural network architecture). I assume that in the evaluation the academic and industrial tools didn’t need to be tuned to the instances.
- The presented approach can only be used to establish that a system satisfies a formal guarantee but not a violation. Model checkers such as ABC and nuXmv establish both. This limitation is not immediately clear from the paper.

**Questions:**

- In theory, the LTL model checking is decidable and has been extensively studied with respect to its computational complexity. Do you expect that any completeness or computational complexity results can be established for the presented approach?
- I am confused about industry tool Y. It is reported even though it does not solve a single instance. Has industry tool Y been developed for these kinds of verification problems?
- Do all benchmarks in the experimental evaluation include a fairness condition?
- I do not understand the relationship that is drawn to reinforcement learning. How is the presented approach related? For example, what would be the MDP?

**Limitations:**

- The paper is transparent about the SMT checks being a bottleneck of the approach.
- The tuning of the approach to individual benchmarks could be discussed in more detail in the limitations section.
- I agree with the authors that no negative societal impacts are expected.

---

> ### Author Rebuttal · Authors · 2024-08-07
>
> * LTL model checking of hardware designs is indeed decidable and PSPACE-complete. While it is theoretically possible to achieve completeness by enumerating all transitions and employing a sufficiently large neural network as a look-up table for the entire state space, this approach is impractical for all but toy examples. We will note this in the final version of our paper.
>
> * Tool Y, a widely used industry tool for formal hardware verification, accepts specifications in SystemVerilog Assertions and thus supports our specifications out of the box. As the tool is proprietary, its internal workings are not disclosed to us, and neither are the reasons for its failure across all instances. We hypothesize that its algorithm may not be optimised for liveness and fairness checking. Consequently, we consider Tool X to be the state-of-the-art for LTL model checking.
>
> * We confirm that all benchmarks include fairness conditions, with further details provided in the appendix.
>
> * Our models are non-deterministic transition systems without probabilities and are hence not MDPs. While our approach shares similarities with reinforcement learning—primarily due to its training on traces or episodes rather than an external dataset—we acknowledge that this similarity may be confusing. We will clarify this in the final version.
>
> * Our method requires some parameter tuning, which is standard for stochastic gradient descent algorithms. Specifically, only two training parameters—the learning rate and clamp upper-bound—need tuning, both optimized across three predefined values. Additionally, our sampling parameter to determine the skewed distribution is chosen among four values (see lines 250-253). It is worth noting that symbolic algorithms also use parameters, such as BDD ordering heuristics, though these are often automated in established tools. Our parameters are certainly suitable for automated tuning methods.
>
> * Finally, we highlight that the presented approach is specialized to establish whether a property is satisfied, as is standard for verification methods based on proof rules (e.g., Hoare logic). The computation of counterexamples is done by combining the method with a bounded model checker, which is the best-known approach to falsification. Our prototype builds on top of the bounded model checker EBMC and hence already includes this capability.

---

> > ### Comment · Reviewer_2gkW · 2024-08-12
> >
> > I have read the rebuttal and thank the authors for their comments and clarifications.

---

### Official Review · Reviewer_ST48 · 2024-07-12

**Soundness:** 3
**Presentation:** 3
**Contribution:** 3
**Rating:** 5
**Confidence:** 4

**Summary:**

This work incorporates neural networks into the model checking process. Specifically, it learns a neural ranking function on random trajectories of the formal model. The ranking function first achieves zero loss on the training set and then is verified for soundness symbolically using SMT solvers. Evaluated on nine different hardware designs, the proposed approach shows significant improvement over prior works.

**Strengths:**

The paper is easy-to-follow with illustrative examples and well motivated. The domain of hardware verification is very important. Moreover, the approach not only provides significant benefits as demonstrated in the experiments but also maintains the soundness guarantee.

**Weaknesses:**

My main concerns are the following.

### 1. Choice of neural architecture
The architecture and some of the layers in Figure 2 seem carefully designed. What are the reasons for choosing these layers? Did you incorporate knowledge of the evaluated tasks into the design of the neural network?

### 2. Size of evaluated tasks
How do the chosen tasks reflect real-world usages? As the paper suggests, increasing the network size would slow SMT check (Do other approaches have the same limitation?). If the size of the evaluated tasks is too small compared to real-world usages, the practical effectiveness of the proposed approach would be rather limited.

### 3. Comparison
The evaluation compares the neural model checker with other checkers as a whole. While this comparison is valuable, there are many differing factors that can influence the results. I suggest adding ablation studies where different algorithms for constructing the ranking functions while keeping the overall system consistent. This would provide a clearer understanding of the algorithmic contribution of the paper.

### 4. Other smaller issues
- At Line 130, the signature of the network contains the parameters of the network. However, at Line 160, the signature does not. I suggest keeping this consistent.
- The tool Industry Y failed on all tasks. What is the reason for this?

**Questions:**

Please consider addressing the points raised in the “Weakness” section.

**Limitations:**

The paper has sufficiently addressed the points concerning limitations and societal impact.

---

> ### Author Rebuttal · Authors · 2024-08-07
>
> 1. Our architecture comprises three main components. The first component is a non-trainable element-wise multiplication layer, designed to normalize the input.  The second component is a trainable element-wise multiplication layer, whose purpose is to automatically focus attention on those inputs that contribute to the ranking. The third component is a trainable, fully connected network that represents the ranking. We employ the clamp activation function, which is standard for quantized neural networks. Overall, our architecture is tailored to address the problem domain effectively. Rather than tuning the architecture for each individual problem instance, we have developed a general architecture that is suitable for all our problems. Details of our architecture are further elaborated in lines 159-173 of the paper.
>
> 2. We utilize a neural network with a consistent number of neurons across all our verification tasks. Consequently, in our experiments, the performance of the SMT check is influenced by the complexity of the overall verification task rather than the number of neurons. Our chosen tasks are based on established design patterns from standard literature, reflecting real-world hardware designs, and we have demonstrated favourable comparisons with both academic and industrial model checkers. Addressing scalability for large hardware designs is a significant challenge for the entire formal verification community, and our results represent a notable step forward in the field.
>
> 3. We appreciate the reviewer's suggestion to conduct an ablation study. In the final version, we will include ablation studies in the appendix to justify our architectural choices. We will measure the performance of our method without the normalisation layer and without the element-wise multiplication layer, as well as different values for the clamp upper bound. We performed a preliminary ablation study on a subset of examples which confirms, as expected, that our normalisation layer and our choice of clamp upper bound are essential for the efficacy of our method. A systematic ablation will indeed make our evaluation even more solid.
>
> 4. As per the reviewer’s suggestion, we will ensure the consistency of the signature of V throughout the paper. Finally, we remark that the internals of industry tool Y are proprietary and not disclosed to us. Although the tool accepts our specifications, we conjecture that its algorithm has not been optimized for full LTL verification. We consider industry tool X to be the leading tool for this purpose.

---

> > ### Comment · Reviewer_ST48 · 2024-08-08
> > **Thanks for the rebuttal**
> >
> > Thanks for submitting the rebuttal. I have read it and will keep the score. I hope the authors incorporate the reviews and the rebuttal into the next version of the paper.

---

### Author Rebuttal · Authors · 2024-08-07

We thank the reviewers for their comments, suggestions and questions, which we will address in the final version as we discuss in this rebuttal. We stress the novelty of our contribution, which introduces a new approach to model-checking temporal logic based on neural certificates. We demonstrated the superior efficacy of our approach compared to the state of the art in hardware model checking on standard hardware designs by means of a direct comparison with both academic and industrial tools. Our procedure guarantees the formal soundness of the result thanks to the combination of neural networks with symbolic reasoning (SMT).

---

### Decision · Program_Chairs · 2024-09-25

**Decision:**

Accept (poster)

**Comment:**

The reviewers were overall positive about the importance of the problem considered, the closeness of the setting to practice, the quality of the presentation, and the improvements in relation to experimental results in the literature.  At the same time, they had some reservations about the choices in the evaluations, the sensitivity of the proposed approach to tuning, and in particular that the limitation to positive instances (as opposed to finding bugs or counterexamples) should be clarified and emphasized.  All four reviews are recommending acceptance, however three of them only at the borderline.